# Calculating Optimistic Likelihoods
# Using (Geodesically) Convex Optimization

**Viet Anh Nguyen**     **Soroosh Shafieezadeh-Abadeh**
École Polytechnique Fédérale de Lausanne, Switzerland
{viet-anh.nguyen, soroosh.shafiee}@epfl.ch

**Man-Chung Yue**
The Hong Kong Polytechnic University, Hong Kong
manchung.yue@polyu.edu.hk

**Daniel Kuhn**
École Polytechnique Fédérale de Lausanne, Switzerland
daniel.kuhn@epfl.ch

**Wolfram Wiesemann**
Imperial College Business School, United Kingdom
ww@imperial.ac.uk

## Abstract

A fundamental problem arising in many areas of machine learning is the evaluation
of the likelihood of a given observation under different nominal distributions.
Frequently, these nominal distributions are themselves estimated from data, which
makes them susceptible to estimation errors. We thus propose to replace each
nominal distribution with an ambiguity set containing all distributions in its vicinity
and to evaluate an *optimistic likelihood*, that is, the maximum of the likelihood
over all distributions in the ambiguity set. When the proximity of distributions
is quantified by the Fisher-Rao distance or the Kullback-Leibler divergence, the
emerging optimistic likelihoods can be computed efficiently using either geodesic
or standard convex optimization techniques. We showcase the advantages of
working with optimistic likelihoods on a classification problem using synthetic as
well as empirical data.

## 1   Introduction

Assume that a set of i.i.d. data points $x_1^M \triangleq x_1, \ldots, x_M \in \mathbb{R}^n$ is generated from one of several
Gaussian distributions $\mathbb{P}_c$, $c \in \mathcal{C}$ with $|\mathcal{C}| < \infty$. To determine the distribution $\mathbb{P}_{c^\star}$, $c^\star \in \mathcal{C}$, under
which $x_1^M$ has the highest likelihood $\ell(x_1^M, \mathbb{P}_c)$ across all $\mathbb{P}_c$, $c \in \mathcal{C}$, we can solve the problem

$$c^\star \in \arg\max_{c \in \mathcal{C}} \left\{ \ell(x_1^M, \mathbb{P}_c) \triangleq -\frac{1}{M} \sum_{m=1}^{M} (x_m - \mu_c)^\top \Sigma_c^{-1} (x_m - \mu_c) - \log \det \Sigma_c \right\}, \quad (1)$$

where $\mu_c$ and $\Sigma_c$ denote the means and covariance matrices that unambiguously characterize the
distributions $\mathbb{P}_c$, $c \in \mathcal{C}$, and the log-likelihood function $\ell(x_1^M, \mathbb{P}_c)$ quantifies the (logarithm of the)
relative probability of observing $x_1^M$ under the Gaussian distribution $\mathbb{P}_c$. Problem (1) naturally
arises in various machine learning applications. In quadratic discriminant analysis, for example,

$x_1^M$ denotes the input values of data samples whose categorical outputs $y_1, \ldots, y_M \in \mathcal{C}$ are to be predicted based on the class-conditional distributions $\mathbb{P}_c$, $c \in \mathcal{C}$ [25]. Likewise, in Bayesian inference with synthetic likelihoods, a Bayesian belief about the models $\mathbb{P}_c$, $c \in \mathcal{C}$, assumed to be Gaussian for computational tractability, is performed based on an observation $x_1^M$ [31, 41]. Problem (1) also arises in likelihood-ratio tests where the null hypothesis '$x_1^M$ is generated by a distribution $\mathbb{P}_c$, $c \in \mathcal{C}_0$' is compared with the alternative hypothesis '$x_1^M$ is generated by a distribution $\mathbb{P}_c$, $c \in \mathcal{C}_1$' [17, 18].

In practice, the parameters $(\mu_c, \Sigma_c)$ of the candidate distributions $\mathbb{P}_c$, $c \in \mathcal{C}$, are typically not known and need to be estimated from data. In quadratic discriminant analysis, for example, it is common to replace the means $\mu_c$ and covariance matrices $\Sigma_c$ with their empirical counterparts $\hat{\mu}_c$ and $\hat{\Sigma}_c$ that are estimated from data. Similarly, the rival model distributions $\mathbb{P}_c$, $c \in \mathcal{C}$, in Bayesian inference with synthetic likelihoods are Gaussian estimates derived from (costly) sampling processes. Unfortunately, problem (1) is highly sensitive to misspecification of the candidate distributions $\mathbb{P}_c$. To combat this problem, we propose to replace the likelihood function in (1) with the *optimistic likelihood*

$$\max_{\mathbb{P} \in \mathcal{P}_c} \ell(x_1^M, \mathbb{P}) \quad \text{with} \quad \mathcal{P}_c = \left\{ \mathbb{P} \in \mathcal{M} : \varphi(\hat{\mathbb{P}}_c, \mathbb{P}) \leq \rho_c \right\}, \tag{2}$$

where $\mathcal{M}$ is the set of all non-degenerate Gaussian distributions on $\mathbb{R}^n$, $\varphi$ is a dissimilarity measure satisfying $\varphi(\mathbb{P}, \mathbb{P}) = 0$ for all $\mathbb{P} \in \mathcal{M}$, and $\rho_c \in \mathbb{R}_+$ are the radii of the ambiguity sets $\mathcal{P}_c$. Problem (2) assumes that the true candidate distributions $\mathbb{P}_c$ are unknown but close to the nominal distributions $\hat{\mathbb{P}}_c$ that are estimated from the training data. In contrast to the log-likelihood $\ell(x_1^M, \mathbb{P}_c)$ that is maximized in problem (1), the optimistic likelihood (2) is of interest in its own right. A common problem in constrained likelihood estimation, for example, is to determine a Gaussian distribution $\mathbb{P}^\star \sim (\mu^\star, \Sigma^\star)$ that is close to a nominal distribution $\mathbb{P}^0 \sim (\mu^0, \Sigma^0)$ reflecting the available prior information such that $x_1^M$ has high likelihood under $\mathbb{P}^\star$ [34]. This task is an instance of the optimistic likelihood evaluation problem (2) with a suitably chosen dissimilarity measure $\varphi$.

Of crucial importance in the generalized likelihood problem (2) is the choice of the dissimilarity measure $\varphi$ as it impacts both the statistical properties as well as the computational complexity of the estimation procedure. A natural choice appears to be the Wasserstein distance, which has recently been popularized in the field of optimal transport [37, 40]. The Wasserstein distance on the space of Gaussian distributions is a Riemannian distance, that is, the distance corresponding the curvilinear geometry on the set of Gaussian distributions induced by the Wasserstein distance, as opposed to the usual distance obtained by treating it as a subset of the space of symmetric matrices. However, since the Wasserstein manifold has a non-negative sectional curvature [37], calculating the associated optimistic likelihood (2) appears to be computationally intractable. Instead, we study the optimistic likelihood under the Fisher-Rao (FR) distance, which is commonly used in signal and image processing [30, 2] as well as computer vision [21, 39]. The FR distance is also a Riemannian metric, and it enjoys many attractive statistical properties that we review in Section 2 of this paper. Most importantly, the FR distance has a non-positive sectional curvature, which implies that the optimistic likelihood (2) reduces to the solution of a geodesically convex optimization problem that is amenable to an efficient solution [6, 35, 38, 42, 43, 44]. We also study problem (2) under the Kullback–Leibler (KL) divergence (or relative entropy), which is intimately related to the FR metric. While the KL divergence lacks some of the desirable statistical features of the FR metric, we will show that it gives rise to optimistic likelihoods that can be evaluated in quasi-closed form by reduction to a one dimensional problem.

While this paper focuses on the parametric approximation of the likelihood where $\mathbb{P}$ belongs to the family of Gaussian distributions, we emphasize that the optimistic likelihood approach can also be utilized in a *non*-parametric setting [28].

The contributions of this paper may be summarized as follows.

1. We show that for Fisher-Rao ambiguity sets, the optimistic likelihood (2) reduces to a geodesically convex problem and is hence amenable to an efficient solution via a Riemannian gradient descent algorithm. We analyze the optimality as well as the convergence of the resulting algorithm.

2. We show that for Kullback-Leibler ambiguity sets, the optimistic likelihood (2) can be evaluated in quasi-closed form by reduction to a one dimensional convex optimization problem.

3. We evaluate the numerical performance of our optimistic likelihoods on a classification problem with artificially generated as well as standard benchmark instances.

Our optimistic likelihoods follow a broader optimization paradigm that exercises optimism in the face of ambiguity. This strategy has been shown to perform well, among others, in multi-armed bandit problems and Bayesian optimization, where the Upper Confidence Bound algorithm takes decisions based on optimistic estimates of the reward [12, 13, 26, 36]. Optimistic optimization has also been successfully applied in support vector machines [8], and it closely relates to sparsity inducing non-convex regularization schemes [29].

The remainder of the paper proceeds as follows. We study the optimistic likelihood (2) under FR and KL ambiguity sets in Sections 2 and 3, respectively. We test our theoretical findings in the context of a classification problem, and we report on numerical experiments in Section 4. Supplementary material and all proofs are provided in the online companion.

**Notation.** Throughout this paper, $\mathbb{S}^n$, $\mathbb{S}^n_+$ and $\mathbb{S}^n_{++}$ denote the spaces of $n$-dimensional symmetric, symmetric positive semi-definite and symmetric positive definite matrices, respectively. For any $A \in \mathbb{R}^{n \times n}$, the trace of $A$ is defined as $\text{Tr}\,(A) = \sum_{i=1}^n A_{ii}$. For any $A \in \mathbb{S}^n$, $\lambda_{\min}(A)$ and $\lambda_{\max}(A)$ denote the minimum and maximum eigenvalues of $A$, respectively. The base of $\log(\cdot)$ is $e$.

## 2 Optimistic Likelihood Problems under the FR Distance

Consider a family of distributions with density functions $p_\theta(x)$, where the parameter $\theta$ ranges over a finite-dimensional smooth manifold $\Theta$. At each point $\theta \in \Theta$, the Fisher information matrix $I_\theta = \mathbb{E}_x[\nabla_\theta \log(p_\theta(x))\nabla_\theta \log(p_\theta(x))^\top | \theta]$ defines an inner product $\langle \cdot, \cdot \rangle_\theta$ on the tangent space $T_\theta\Theta$ by $\langle \zeta_1, \zeta_2 \rangle_\theta = \zeta_1^T I_\theta \zeta_2$ for $\zeta_1, \zeta_2 \in T_\theta\Theta$. The family of inner products $\{\langle \cdot, \cdot \rangle_\theta\}_{\theta \in \Theta}$ on the tangent spaces then defines a Riemannian metric, called the FR metric. The FR distance on $\Theta$ is the geodesic distance associated with the FR metric, *i.e.*, the FR distance between the two points $\theta_0, \theta_1 \in \Theta$ is

$$d(\theta_0, \theta_1) = \inf_\gamma \int_0^1 \sqrt{\langle \gamma'(t), \gamma'(t) \rangle_{\gamma(t)}} dt,$$

where the infimum is taken over all smooth curves $\gamma : [0, 1] \to \Theta$ with $\gamma(0) = \theta_0$ and $\gamma(1) = \theta_1$. Any curve $\gamma$ attaining the infimum is said to be a geodesic from $\theta_0$ to $\theta_1$. The FR metric represents a natural distance measure for parametric families of probability distributions as it is invariant under transformations on the data space (the $x$ space) by a class of statistically important mappings, and it is the unique (up to a scaling) Riemannian metric enjoying such a property, see [14, 15, 5].

Since the covariance matrix is more difficult to estimate than the mean (see Appendix A), we focus here on the family of all Gaussian distributions with a fixed mean vector $\hat{\mu} \in \mathbb{R}^n$. These distributions are parameterized by $\theta = \Sigma$, that is, the covariance matrix. The parameter manifold is thus given by $\Theta = \mathbb{S}^n_{++}$. On this manifold, the FR distance is available in closed form.[1]

**Proposition 2.1** (FR distance for Gaussian distributions [3])**.** If $\mathcal{N}(\hat{\mu}, \Sigma_0)$ and $\mathcal{N}(\hat{\mu}, \Sigma_1)$ are Gaussian distributions with identical mean $\hat{\mu} \in \mathbb{R}^n$ and covariance matrices $\Sigma_0, \Sigma_1 \in \mathbb{S}^n_{++}$, we have

$$d(\Sigma_0, \Sigma_1) = \frac{1}{\sqrt{2}} \left\| \log(\Sigma_1^{-\frac{1}{2}} \Sigma_0 \Sigma_1^{-\frac{1}{2}}) \right\|_F, \tag{3}$$

where $\log(\cdot)$ represents the matrix logarithm, and $\|\cdot\|_F$ stands for the Frobenius norm.

The distance $d(\cdot, \cdot)$ is invariant under *inversions* and *congruent transformations* of the input parameters [32, Proposition 1], *i.e.*, for any $\hat{\Sigma}, \Sigma \in \mathbb{S}^n_{++}$ and invertible matrix $A \in \mathbb{R}^{n \times n}$, we have

$$d(\hat{\Sigma}^{-1}, \Sigma^{-1}) = d(\hat{\Sigma}, \Sigma) \tag{4}$$

$$\text{and} \quad d(A\hat{\Sigma}A^\top, A\Sigma A^\top) = d(\hat{\Sigma}, \Sigma). \tag{5}$$

By the inversion invariance (4), the distance $d(\cdot, \cdot)$ is independent of whether we use the covariance matrix $\Sigma$ or the precision matrix $\Sigma^{-1}$ to parametrize normal distributions. Note that if $x_1 \sim \mathcal{N}(\mu, \Sigma_1)$ and $x_2 \sim \mathcal{N}(\mu, \Sigma_2)$, then $Ax_1 + b \sim \mathcal{N}(A\mu + b, A\Sigma_1 A^\top)$ and $Ax_2 + b \sim \mathcal{N}(A\mu + b, A\Sigma_2 A^\top)$. By the congruence invariance (5), the distance $d(\cdot, \cdot)$ thus remains unchanged under affine transformations

$x \to Ax + b$. Remarkably, the invariance property (5) uniquely characterizes the distance $d(\cdot, \cdot)$. More precisely, any Riemannian distance satisfying the invariance property (5) coincides (up to a scaling) with the distance $d(\cdot, \cdot)$, see, for example, [33, Section 3] and [9, Section 2].

We now study the optimistic likelihood problem (2), where the FR distance is used as the dissimilarity measure. Given a data batch $x_1^M$ and a radius $\rho > 0$, the optimistic likelihood problem reduces to

$$\min_{\Sigma \in \mathcal{B}^{\mathrm{FR}}} L(\Sigma), \quad \text{where} \quad \begin{cases} L(\Sigma) \triangleq \langle S, \Sigma^{-1} \rangle + \log \det \Sigma, \\ \mathcal{B}^{\mathrm{FR}} \triangleq \{\Sigma \in \mathbb{S}_{++}^n : d(\Sigma, \hat{\Sigma}) \le \rho\}, \end{cases} \tag{6}$$

and $S = M^{-1} \sum_{m=1}^M (x_m - \hat{\mu})(x_m - \hat{\mu})^\top$ stands for the sample covariance matrix.

We next prove that problem (6) is solvable, which justifies the use of the minimization operator.

**Lemma 2.2.** *The optimal value of problem (6) is finite and is attained by some $\Sigma^\star \in \mathcal{B}^{\mathrm{FR}}$.*

Even though the objective function of (6) involves a concave log-det term, it can be shown to be convex over the region $0 \prec \Sigma \preceq 2S$ [10, Exercise 7.4]. However, in practice $S$ may be singular, in which case this region becomes empty. Maximum likelihood estimation problems akin to (6) are often reparameterized in terms of the precision matrix $X = \Sigma^{-1}$. In this case, (6) becomes

$$\min \left\{ \langle S, X \rangle - \log \det X : X \in \mathbb{S}_{++}^n, \| \log(X^{\frac{1}{2}} \hat{\Sigma} X^{\frac{1}{2}}) \|_F \le \sqrt{2}\rho \right\}.$$

Even though this reparameterization convexifies the objective, it renders the feasible set non-convex.

## 2.1 Geodesic Convexity of the Optimistic Likelihood Problem

As problem (6) cannot be addressed with standard methods from convex optimization, we re-interpret it as a constrained minimization problem on the Riemannian manifold $\Theta = \mathbb{S}_{++}^n$ endowed with the FR metric. The key advantage of this approach is that we can show problem (6) to be *geodesically convex*. Geodesic convexity generalizes the usual notion of convexity in Euclidean spaces to Riemannian manifolds. We can thus solve problem (6) via algorithms from geodesically convex optimization, which inherit many benefits of the standard algorithms of convex optimization in Euclidean spaces.

The Riemannian manifold $\Theta = \mathbb{S}_{++}^n$ endowed with the FR metric is in fact a Hadamard manifold, that is, a complete simply connected Riemannian manifold with non-positive sectional curvature, see [22, Theorem XII 1.2]. Thus, any two points are connected by a *unique* geodesic [11]. By [7, Theorem 6.1.6], for $\Sigma_0, \Sigma_1 \in \mathbb{S}_{++}^n$, the unique geodesic $\gamma : [0, 1] \to \mathbb{S}_{++}^n$ from $\Sigma_0$ to $\Sigma_1$ is given by

$$\gamma(t) = \Sigma_0^{\frac{1}{2}} \left( \Sigma_0^{-\frac{1}{2}} \Sigma_1 \Sigma_0^{-\frac{1}{2}} \right)^t \Sigma_0^{\frac{1}{2}}. \tag{7}$$

We are now ready to give precise definitions of geodesically convex sets and functions on Hadamard manifolds. We emphasize that these definitions would be more subtle for general Riemannian manifolds, which can have several geodesics between two points.

**Definition 2.3** (Geodesically convex set). *A set $\mathcal{U} \subseteq \mathbb{S}_{++}^n$ is said to be geodesically convex if for all $\Sigma_0, \Sigma_1 \in \mathcal{U}$, the image of the unique geodesic from $\Sigma_0$ to $\Sigma_1$ is contained in $\mathcal{U}$, i.e., $\gamma([0, 1]) \subseteq \mathcal{U}$.*

**Definition 2.4** (Geodesically convex function). *A function $f : \mathbb{S}_{++}^n \to \mathbb{R}$ is said to be geodesically convex if for all $\Sigma_0, \Sigma_1 \in \mathbb{S}_{++}^n$, the unique geodesic $\gamma$ from $\Sigma_0$ to $\Sigma_1$ satisfies $f(\gamma(t)) \le (1 - t)f(\Sigma_0) + tf(\Sigma_1) \, \forall t \in [0, 1]$.*

In order to prove that (6) is a geodesically convex optimization problem, we need to establish the geodesic convexity of the feasible region $\mathcal{B}^{\mathrm{FR}}$ and the loss function $L(\cdot)$. Note that, in stark contrast to Euclidean geometry, a geodesic ball on a general manifold may not be geodesically convex.[2]

**Theorem 2.5** (Geodesic convexity of problem (6)). *For any $\hat{\Sigma} \in \mathbb{S}_{++}^n$, $S \in \mathbb{S}_+^n$ and $\rho \in \mathbb{R}_+$, $\mathcal{B}^{\mathrm{FR}}$ is a geodesically convex set, and $L(\cdot)$ is a geodesically convex function over $\mathbb{S}_{++}^n$.*

Theorem 2.5 establishes that the optimistic likelihood problem (6), which is non-convex with respect to the usual Euclidean geometry on the embedding space $\mathbb{R}^{n \times n}$, is actually convex with respect to the Riemannian geometry on $\mathbb{S}_{++}^n$ induced by the FR metric.

**Algorithm 1** Projected Geodesic Gradient Descent Algorithm

---

**Input:** $\hat{\Sigma} \succ 0$, $\rho > 0$, $S \succeq 0$, $K \in \mathbb{N}$, $\{\alpha_k\}_{k=1}^{K} \subseteq \mathbb{R}_{++}$
**Initialization:** Set $\Sigma_1 \leftarrow \hat{\Sigma}$, $\bar{\Sigma}_1 \leftarrow \hat{\Sigma}$
**for** $k = 1, 2, \ldots, K-1$ **do**
    Compute the Riemannian gradient at $\Sigma_k$: $G_k \leftarrow 2(\Sigma_k - S)$
    Perform a gradient descent step using the exponential map:

$$\Sigma_{k+\frac{1}{2}} \leftarrow \mathrm{Exp}_{\Sigma_k}(-\alpha_k G_k) = \Sigma_k^{\frac{1}{2}} \exp\left(-\alpha_k \Sigma_k^{-\frac{1}{2}} G_k \Sigma_k^{-\frac{1}{2}}\right) \Sigma_k^{\frac{1}{2}}$$

    Project $\Sigma_{k+\frac{1}{2}}$ onto $\mathcal{B}^{\mathrm{FR}}$: $\Sigma_{k+1} \leftarrow \mathrm{Proj}_{\mathcal{B}^{\mathrm{FR}}}(\Sigma_{k+\frac{1}{2}})$
    Compute the new iterate by interpolation: $\bar{\Sigma}_{k+1} \leftarrow \mathrm{Exp}_{\bar{\Sigma}_k}\left(\frac{1}{k+1} \mathrm{Exp}_{\bar{\Sigma}_k}^{-1}(\Sigma_{k+1})\right)$
**end for**
**Output:** Report the last iterate $\bar{\Sigma}_K$ as an approximate solution

---

## 2.2 Projected Geodesic Gradient Descent Algorithm

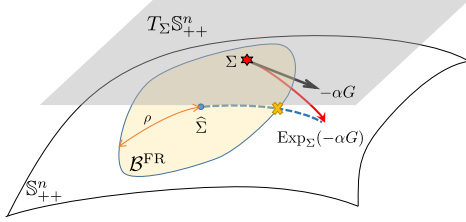

Figure 1: Visualization of the FR ball $\mathcal{B}^{\mathrm{FR}}$ (yellow set) within the manifold $\mathbb{S}_{++}^n$ (white set).

In the same way as the convexity of a standard constrained optimization problem can be exploited to find a global minimizer via a projected gradient descent algorithm, the geodesic convexity of problem (6) can be exploited to find a global minimizer by using a *projected geodesic gradient descent* algorithm of the type described in [43]. The mechanics of a generic iteration are visualized in Figure 1. As in any gradient descent method, given the current iterate $\Sigma$, we first need to compute the direction along which the objective function $L$ decreases fastest. In the context of optimization on manifolds, this direction corresponds to the negative Riemannian gradient $-G$ at point $\Sigma$, which belongs to the tangent space $T_\Sigma \mathbb{S}_{++}^n \simeq \mathbb{S}^n$. Unfortunately, the curve $\gamma(\alpha) = \Sigma - \alpha G$ fails to be a geodesic and will eventually leave the manifold for sufficiently large step sizes $\alpha$. This prompts us to construct the (unique) geodesic that emanates from point $\Sigma$ with initial velocity $-G$. Formally, this geodesic can be represented as $\gamma(\alpha) = \mathrm{Exp}_\Sigma(-\alpha G)$, where $\mathrm{Exp}_\Sigma(\cdot)$ denotes the *exponential map* at $\Sigma$. As we will see below, this geodesic (visualized by the red curve) remains within the manifold for any $\alpha > 0$ but may eventually leave the feasible region $\mathcal{B}^{\mathrm{FR}}$. If this happens for the chosen step size $\alpha$, we project $\mathrm{Exp}_\Sigma(-\alpha G)$ back onto the feasible region, that is, we map it to its closest point in $\mathcal{B}^{\mathrm{FR}}$ with respect to the FR distance (visualized by the yellow cross). Denoting this FR projection by $\mathrm{Proj}_{\mathcal{B}^{\mathrm{FR}}}(\cdot)$, the next iterate of the projected geodesic gradient descent algorithm can thus be expressed as $\Sigma^+ = \mathrm{Proj}_{\mathcal{B}^{\mathrm{FR}}}(\mathrm{Exp}_\Sigma(-\alpha G))$.

Starting from $\Sigma_1 = \hat{\Sigma}$, the proposed algorithm constructs $K$ iterates $\{\Sigma_k\}_{k=1}^{K}$ via the above recursion. As in [43], the algorithm also constructs a second sequence $\{\bar{\Sigma}_k\}_{k=1}^{K}$ of feasible covariance matrices with $\bar{\Sigma}_1 = \hat{\Sigma}$ and $\bar{\Sigma}_{k+1} = \bar{\gamma}(1/(k+1))$ for $k = 1, \ldots, K-1$, where $\bar{\gamma}(t)$ represents the geodesic (7) connecting $\bar{\Sigma}_k$ with $\Sigma_{k+1}$. Thus, $\bar{\Sigma}_{k+1}$ is defined as a *geodesic convex combination* of $\bar{\Sigma}_k$ and $\Sigma_{k+1}$. A precise description of the proposed algorithm in pseudocode is provided in Algorithm 1.

In the following we show that the Riemannian gradient, the exponential map $\mathrm{Exp}_\Sigma(\cdot)$ as well as the projection $\mathrm{Proj}_{\mathcal{B}^{\mathrm{FR}}}(\cdot)$ can all be evaluated in closed form in $\mathcal{O}(n^3)$.

By [3, Page 362], the FR metric on the tangent space $T_\Sigma \mathbb{S}_{++}^n$ at $\Sigma \in \mathbb{S}_{++}^n$ can be re-expressed as

$$\langle \Omega_1, \Omega_2 \rangle_\Sigma \triangleq \frac{1}{2} \mathrm{Tr}\left(\Omega_1 \Sigma^{-1} \Omega_2 \Sigma^{-1}\right) \quad \forall \Omega_1, \Omega_2 \in T_\Sigma \mathbb{S}_{++}^n. \tag{8}$$

Using (8) and [1, Equation 3.32], the Riemannian gradient $G = \mathrm{grad}\, L$ of the objective function $L(\cdot)$ at $\Sigma$ can be computed from the Euclidean gradient $\nabla L(\Sigma)$ as

$$\mathrm{grad}\, L(\Sigma) = 2\Sigma(\nabla L(\Sigma))\Sigma = 2\Sigma(\Sigma^{-1} - \Sigma^{-1}S\Sigma^{-1})\Sigma = 2(\Sigma - S). \tag{9}$$

Moreover, by [35, Equation (3.2)], the exponential map $\mathrm{Exp}_\Sigma : T_\Sigma \mathbb{S}^n_{++} \to \mathbb{S}^n_{++}$ at $\Sigma$ is given by

$$\mathrm{Exp}_\Sigma(G) = \Sigma^{\frac{1}{2}} \exp(\Sigma^{-\frac{1}{2}} G \Sigma^{-\frac{1}{2}}) \Sigma^{\frac{1}{2}}, \quad G \in T_\Sigma \mathbb{S}^n_{++} \simeq \mathbb{S}^n,$$

where $\exp(\cdot)$ denotes the matrix exponential. The inverse map $\mathrm{Exp}_\Sigma^{-1} : \mathbb{S}^n_{++} \to T_\Sigma \mathbb{S}^n_{++}$ satisfies

$$\mathrm{Exp}_\Sigma^{-1}(A) = \Sigma^{\frac{1}{2}} \big( \log \Sigma^{-\frac{1}{2}} A \Sigma^{-\frac{1}{2}} \big) \Sigma^{\frac{1}{2}}, \quad A \in \mathbb{S}^n_{++}.$$

Finally, the projection $\mathrm{Proj}_\mathcal{B}(\cdot)$ onto $\mathcal{B}^{\mathrm{FR}}$ with respect to the FR distance is defined through

$$\mathrm{Proj}_{\mathcal{B}^{\mathrm{FR}}}(\Sigma') \triangleq \arg\min_{\Sigma \in \mathcal{B}^{\mathrm{FR}}} d(\Sigma, \Sigma'), \quad \Sigma' \in \mathbb{S}^n_{++}. \tag{10}$$

The following lemma ensures that this projection is well-defined and admits a closed-form expression.

**Lemma 2.6** (Projection onto $\mathcal{B}^{\mathrm{FR}}$)**.** For any $\Sigma' \in \mathbb{S}^n_{++}$ with $d(\hat{\Sigma}, \Sigma') = \rho'$ the following hold.

(i) There $\arg\min$-mapping in (10) is a singleton, and thus $\mathrm{Proj}_{\mathcal{B}^{\mathrm{FR}}}(\Sigma')$ is well-defined.

(ii) The projection of $\Sigma'$ onto $\mathcal{B}^{\mathrm{FR}}$ is given by

$$\mathrm{Proj}_{\mathcal{B}^{\mathrm{FR}}}(\Sigma') = \begin{cases} \hat{\Sigma}^{\frac{1}{2}} \big( \hat{\Sigma}^{-\frac{1}{2}} \Sigma' \hat{\Sigma}^{-\frac{1}{2}} \big)^{\frac{\rho}{\rho'}} \hat{\Sigma}^{\frac{1}{2}} & \text{if } \rho' > \rho, \\ \Sigma' & \text{otherwise.} \end{cases} \tag{11}$$

By comparison with (7), one easily verifies that $\mathrm{Proj}_{\mathcal{B}^{\mathrm{FR}}}(\Sigma')$ constitutes a geodesic convex combination between $\Sigma'$ and $\hat{\Sigma}$. Figure 1 visualizes the geodesic from $\hat{\Sigma}$ to $\Sigma'$ by the blue dashed line. Therefore, the projection $\mathrm{Proj}_{\mathcal{B}^{\mathrm{FR}}}$ onto the FR ball $\mathcal{B}^{\mathrm{FR}}$ within $\mathbb{S}^n_{++}$ endowed with the FR metric is constructed in a similar manner as the projection onto a Euclidean ball within a Euclidean space.

The following theorem asserts that Algorithm 1 enjoys a sublinear convergence rate.

**Theorem 2.7** (Sublinear convergence rate)**.** With a constant stepsize

$$\alpha_k \equiv 2^{1/4} \sqrt{\rho \tanh(2\sqrt{2}\rho)} / (\Gamma \sqrt{K}),$$

where $\Gamma \triangleq 2^{-1/2} \sqrt{n} \cdot e^{2\sqrt{2}\rho} \cdot \lambda_{\min}^{-2}(\hat{\Sigma}) \cdot \max\{|1 - e^{\sqrt{2}\rho} \lambda_{\min}^{-1}(\hat{\Sigma}) \lambda_{\max}(S)|, 1\}$, Algorithm 1 satisfies

$$L(\bar{\Sigma}_K) - L(\Sigma^\star) \leq \frac{2^{\frac{7}{4}} \rho^{\frac{3}{2}} \Gamma}{\sqrt{K \tanh(2\sqrt{2}\rho)}}.$$

The proof of Theorem 2.7 closely follows that of [43, Theorem 9]. The difference is that [43, Theorem 9] requires the objective function to be Lipschitz continuous on $\mathbb{S}^n_{++}$. Unfortunately, such an assumption is not satisfied by $L(\cdot)$. We circumvent this by proving that the Riemannian gradient of $L(\cdot)$ is bounded uniformly on $\mathcal{B}^{\mathrm{FR}}$.

Endeavors are currently underway to devise algorithms for minimizing a geodesically strongly convex objective function over a geodesically convex feasible set that offer a linear convergence guarantee, see, e.g., [43, Theorem 15]. The next lemma shows that the objective function of problem (6) is indeed geodesically smooth and geodesically strongly convex[3] whenever $S \succ 0$. This suggests that the empirical performance of Algorithm 1 could be significantly better than the theoretical guarantee of Theorem 2.7. Indeed, our numerical results in Section 4.1 confirm that if $S \succ 0$, then Algorithm 1 displays a linear convergence rate.

**Lemma 2.8** (Strong convexity and smoothness of $L(\cdot)$)**.** The objective function $L(\cdot)$ of problem (6) is geodesically $\beta$-smooth on $\mathcal{B}^{\mathrm{FR}}$ with

$$\beta = \frac{2\lambda_{\max}(S)}{\lambda_{\min}(\hat{\Sigma}) \exp(-\sqrt{2}\rho)}.$$

If $S \succ 0$, then $L(\cdot)$ is also geodesically $\sigma$-strongly convex on $\mathcal{B}^{\mathrm{FR}}$ with

$$\sigma = \frac{2\lambda_{\min}(S)}{\lambda_{\max}(\hat{\Sigma}) \exp(\sqrt{2}\rho)}.$$

**Remark 2.9.** Problem (6) could also be addressed with the algorithmic framework developed in [24]. Due to space limitations, we leave this for future research.

# 3 Generalized Likelihood Estimation under the KL Divergence

The KL divergence, which is widely used in information theory [16, § 2], can be employed as an alternative dissimilarity measure in the optimistic likelihood problem (2). If both $\hat{\mathbb{P}}$ and $\mathbb{P}$ are Gaussian distributions, then the KL divergence from $\hat{\mathbb{P}}$ to $\mathbb{P}$ admits an analytical expression.

**Proposition 3.1** (KL divergence for Gaussian distributions). *For any $\hat{\mu} \in \mathbb{R}^n$ and $\Sigma_0, \Sigma_1 \in \mathbb{S}^n_{++}$, the KL divergence from $\mathbb{P}_0 = \mathcal{N}(\hat{\mu}, \Sigma_0)$ to $\mathbb{P}_1 = \mathcal{N}(\hat{\mu}, \Sigma_1)$ amounts to*

$$\mathrm{KL}(\mathbb{P}_0 \parallel \mathbb{P}_1) = \frac{1}{2}\big( \mathrm{Tr}\big( \Sigma_1^{-1}\Sigma_0 \big) + \log \det(\Sigma_1 \Sigma_0^{-1}) - n \big).$$

Unlike the FR distance, the KL divergence is not symmetric. Proposition 3.1 implies that if the KL divergence is used as the dissimilarity measure, then the optimistic likelihood problem (2) reduces to

$$\min_{\Sigma \succ 0} \quad \Big\{ \mathrm{Tr}\big( S\Sigma^{-1} \big) + \log \det \Sigma : \mathrm{Tr}\big( \Sigma^{-1}\hat{\Sigma} \big) + \log \det(\Sigma \hat{\Sigma}^{-1}) - n \leq 2\rho \Big\}, \qquad (12)$$

where $S = M^{-1} \sum_{m=1}^{M} (x_m - \hat{\mu})(x_m - \hat{\mu})^\top$ denotes again the sample covariance matrix. Because of the concave log-det terms in the objective and the constraints, problem (12) is non-convex. By using the variable substitution $X \leftarrow \Sigma^{-1}$, however, problem (12) can be reduced to a univariate convex optimization problem and thereby solved in quasi-closed form.

**Theorem 3.2.** *For any $\hat{\Sigma} \succ 0$ and $\rho > 0$, the optimal value of problem (12) amounts to*

$$(1 + \gamma^\star) \, \mathrm{Tr}\big( S(S + \gamma^\star\hat{\Sigma})^{-1} \big) + \log \det(S + \gamma^\star\hat{\Sigma}) - n \log(1 + \gamma^\star),$$

*where $\gamma^\star$ is the unique optimal solution of the univariate convex optimization problem*

$$\min_{\gamma > 0} \Big\{ \gamma(2\rho + \log \det \hat{\Sigma}) + n(1 + \gamma) \log(1 + \gamma) - (1 + \gamma) \log \det(S + \gamma\hat{\Sigma}) \Big\}. \qquad (13)$$

Problem (13) can be solved efficiently using state-of-the-art first- or second-order methods, see Appendix E. However, in each iteration we still need to evaluate the determinant of a positive definite $n$-by-$n$ matrix, which requires $\mathcal{O}(n^3)$ arithmetic operations. The following corollary shows that this computational burden can be alleviated when the sample covariance matrix $S$ has low rank.

**Corollary 3.3** (Singular sample covariance matrices). *If $S = \Lambda\Lambda^\top$ for some $\Lambda \in \mathbb{R}^{n \times k}$ and $k \in \mathbb{N}$ with $k < n$, then problem (13) simplifies to*

$$\min_{\gamma > 0} \Big\{ 2\gamma\rho + n(1 + \gamma) \log(1 + \gamma) - (n - k)(1 + \gamma) \log \gamma - (1 + \gamma) \log \det(\gamma I_k + \Lambda^\top \hat{\Sigma}^{-1} \Lambda) \Big\}.$$

We will see that for classification problems the matrix $S$ has rank 1, in which case the log-det term in the above univariate convex program reduces to the scalar logarithm. In Appendix E we provide explicit first- and second-order derivatives of the objective of problem (13) and its simplification.

# 4 Numerical Results

We investigate the empirical behavior of our projected geodesic gradient descent algorithm (Section 4.1) and the predictive power of our flexible discriminant rules (Section 4.2). Our algorithm and all tests are implemented in Python, and the source code is available from `https://github.com/sorooshafiee/Optimistic_Likelihoods`.

## 4.1 Convergence Behavior of the Projected Geodesic Descent Algorithm

To study the empirical convergence behavior of Algorithm 1, for $n \in \{10, 20, \ldots, 100\}$ we generate 100 covariance matrices $\hat{\Sigma}$ according to the following procedure. We *(i)* draw a standard normal random matrix $B \in \mathbb{R}^{n \times n}$ and compute $A = B + B^\top$; we *(ii)* conduct an eigenvalue decomposition $A = RDR^T$; we *(iii)* replace $D$ with a random diagonal matrix $\hat{D}$ whose diagonal elements are sampled uniformly from $[1, 10]^n$; and we *(iv)* set $\hat{\Sigma} = R\hat{D}R^\top$. For each of these covariance matrices, we set $\hat{\mu} = 0$, $M = 1$, $x_1^M \triangleq x$ for a standard normal random vector $x \in \mathbb{R}^n$ and calculate

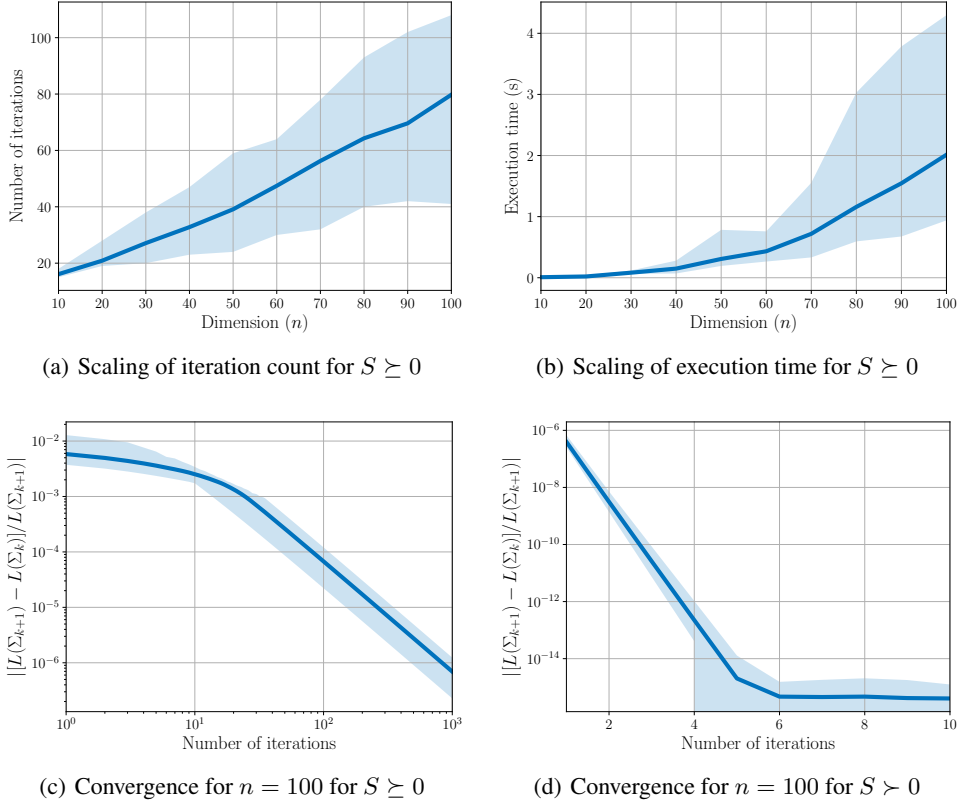

(a) Scaling of iteration count for $S \succeq 0$

(b) Scaling of execution time for $S \succeq 0$

(c) Convergence for $n = 100$ for $S \succeq 0$

(d) Convergence for $n = 100$ for $S \succ 0$

Figure 2: Convergence behavior of the projected geodesic gradient descent algorithm. Solid lines (shaded regions) represent averages (ranges) across 100 independent simulations.

the optimistic likelihood (6) for $\rho = \sqrt{n}/100$. This choice of $\rho$ ensures that the radius of the ambiguity set scales with $n$ in the same way as the Frobenius norm. Figures 2(a) and 2(b) report the number of iterations as well as the overall execution time of Algorithm 1 when we terminate the algorithm as soon as the relative improvement $|[L(\Sigma_{k+1}) - L(\Sigma_k)]/L(\Sigma_{k+1})|$ drops below $0.01\%$. Notice that the number of required iterations scales linearly with $n$ while the overall runtime grows polynomially with $n$. Figure 2(c) shows the relative improvement as a function of the iteration count. Empirically, the number of iterations scales with $\mathcal{O}(1/k^2)$, which is faster than the theoretical rate established in Theorem 2.7. We also study the empirical convergence behavior of Algorithm 1 when the input matrix $S$ is positive definite. We repeat the first experiment with $M = 100$, and we set $S = \delta I + \sum_{i=1}^{M} x_i x_i^\top / M$ for $\delta = 10^{-6}$ to ensure that $S$ is positive definite. Figure 2(d) indicates that, in this case, the empirical convergence rate of Algorithm 1 is linear.

## 4.2 Application: Flexible Discriminant Rules

Consider a classification problem where a categorical response $Y \in \mathcal{C}$, $\mathcal{C} = \{1, \ldots, C\}$, should be predicted from continuous inputs $X \in \mathbb{R}^n$. In this context, Bayes' Theorem implies that $\mathbb{P}(Y = c | X = x) \propto \pi_c \cdot f_c(x)$, $c \in \mathcal{C}$, where $\pi_c = \mathbb{P}(Y = c)$ denotes the prior probability of the response belonging to class $c$, and $f_c$ is the density function of $X$ for an observation of class $c$. In practice, $\pi_c$ and $f_c$ are unknown and need to be estimated from a training data set $(\hat{x}_1, \hat{y}_1), \ldots, (\hat{x}_N, \hat{y}_N)$. Assuming that the densities $f_c$, $c \in \mathcal{C}$, correspond to Gaussian distributions with (unknown) class-specific means $\mu_c$ and covariance matrices $\Sigma_c$, the quadratic discriminant analysis (QDA) replaces $\pi_c$ with $\hat{\pi}_c = N_c/N$, where $N_c = |\{i : \hat{y}_i = c\}|$, and $f_c$ with the density of the Gaussian distribution $\hat{\mathbb{P}}_c \sim \mathcal{N}(\hat{\mu}_c, \hat{\Sigma}_c)$, whose mean and covariance matrix are estimated from the training data, to classify a new observation $x$ using the discriminant rule

$$\mathcal{C}_{\text{QDA}}(x) \in \arg\max_{c \in \mathcal{C}} \left\{ \frac{1}{2} \ell(x, \hat{\mathbb{P}}_c) + \log(\hat{\pi}_c) \right\}.$$

Table 1: Average correct classification rates on the benchmark instances

|  | FQDA | KQDA | QDA | RQDA | SQDA | WQDA |
|---|---|---|---|---|---|---|
| Australian | 80.68 | **83.68** | 80.03 | 79.76 | 80.73 | 79.94 |
| Banknote authentication | 99.07 | **99.47** | 98.56 | 98.54 | 98.53 | 98.54 |
| Climate model | 94.46 | **94.55** | 91.78 | 92.72 | 94.42 | 92.78 |
| Cylinder | 70.69 | 70.67 | 67.10 | 70.33 | **70.99** | 70.34 |
| Diabetic | **75.97** | 74.53 | 74.19 | 74.60 | 74.70 | 75.04 |
| Fourclass | **80.13** | 79.97 | 79.32 | 79.32 | 79.32 | 79.33 |
| German credit | 74.50 | 74.60 | 71.41 | **76.18** | 74.99 | 76.31 |
| Haberman | 74.87 | **75.41** | 74.92 | 74.96 | 75.04 | 74.97 |
| Heart | **84.23** | 83.31 | 81.42 | 82.62 | 84.17 | 82.42 |
| Housing | 88.89 | **92.90** | 88.54 | 87.01 | 81.69 | 88.31 |
| Ilpd | 57.42 | **57.83** | 55.18 | 54.97 | 55.45 | 55.15 |
| Mammographic mass | 80.66 | 80.85 | 80.37 | 80.88 | **81.05** | 80.65 |
| Pima | **75.97** | 74.53 | 74.19 | 74.60 | 74.70 | 75.04 |
| Ringnorm | **98.69** | 98.65 | 98.56 | 98.56 | 98.65 | 98.56 |

Here, the likelihood $\ell(x, \hat{\mathbb{P}}_c)$ is defined as in (1) for $M = 1$. If $\hat{\pi}_1 = \ldots = \hat{\pi}_C$, this classification rule reduces to the maximum likelihood discrimant rule [20, § 14].

QDA can be sensitive to misspecifications of the empirical moments. To reduce this sensitivity, we replace the nominal Gaussian distributions $\hat{\mathbb{P}}_c$ with the Gaussian distributions $\mathbb{P}_c^\star$ that would have generated the sample $x$ with highest likelihood, among all Gaussian distributions in the vicinity of the nominal distributions $\hat{\mathbb{P}}_c$. This results in a *flexible discriminant rule* of the form

$$\mathcal{C}_{\text{flex}}(x) \in \arg\max_{c \in \mathcal{C}} \ \max_{\mathbb{P} \in \mathcal{P}_c} \left\{ \frac{1}{2} \ell(x, \mathbb{P}) + \log(\hat{\pi}_c) \right\},$$

which makes use of the optimistic likelihoods (2). Here, $\mathcal{P}_c$ is the FR or KL ball centered at the nominal distribution $\hat{\mathbb{P}}_c$. To ensure that $\hat{\Sigma}_c \succ 0$ for all $c \in \mathcal{C}$, we use the Ledoit-Wolf covariance estimator [23], which is parameter-free and returns a well-conditioned matrix by minimizing the mean squared error between the estimated and the real covariance matrix.

We compare the performance of our flexible discriminant rules with standard QDA implementations from the literature on datasets from the UCI repository [4]. Specifically, we compare the following methods.

- **FQDA** and **KQDA**: our flexible discriminant rules based on FR (FQDA) and KL (KQDA) ambiguity sets with radii $\rho_c$;
- **QDA**: regular QDA with empirical means and covariance matrices estimated from data;
- **RQDA**: regularized QDA based on the linear shrinkage covariance estimator $\hat{\Sigma}_c + \rho_c I_n$;
- **SQDA**: sparse QDA based on the graphical lasso covariance estimator [19] with parameter $\rho_c$;
- **WQDA**: Wasserstein QDA based on the nonlinear shrinkage approach [27] with parameter $\rho_c$.

All results are averaged across 100 independent trials for $\rho_c \in \{a\sqrt{n} \cdot 10^b : a \in \{1, \ldots, 9\}, \ b \in \{-3, -2, -1\}\}$. In each trial, we randomly select 75% of the data for training and the remaining 25% for testing. The size of the ambiguity set and the regularization parameter are selected using stratified 5-fold cross validation. The performance of the classifiers is measured by the *correct classification rate* (CCR). The average CCR scores over the 100 trials are reported in Table 1.

**Acknowledgments** We gratefully acknowledge financial support from the Swiss National Science Foundation under grant BSCGI0_157733 as well as the EPSRC grants EP/M028240/1, EP/M027856/1 and EP/N020030/1.

## Footnotes

[1]We can also handle the case where the covariance matrix is fixed but the mean is subject to ambiguity, see Appendix B. However, as there is no closed-form expression for the FR distance between two generic Gaussian distributions, we cannot handle the case where both the mean and the covariance matrix are subject to ambiguity.

[2]For example, consider the circle $S^1 \triangleq \{x \in \mathbb{R}^2 : \|x\|_2 = 1\}$ which is a 1-dimensional manifold. Any major arc is a geodesic ball but *not* a geodesically convex subset of $S^1$.

[3] The strong convexity and smoothness properties are defined in Definitions C.4 and C.5, respectively.

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
