[Supplementary Material · QDA_proof.pdf]

# A   Justification for using Ambiguity Sets with a Fixed Mean Vector

We provide some empirical evidence to justify the ambiguity sets with a fixed mean vector used in Sections 2 and 3. Towards this end, we first fix a matrix $A \in \mathbb{R}^{n \times n}$, where each element is drawn independently from a standard Gaussian distribution, and set $\Sigma \triangleq AA^\top$. We then generate $N \in \{20, \dots, 100\}$ i.i.d. samples $\hat{x}_1, \dots, \hat{x}_N$ from the Gaussian distribution $\mathbb{Q} = \mathcal{N}(0, \Sigma)$, and compute the empirical mean $\hat{\mu}_N$ and the empirical covariance matrix $\hat{\Sigma}_N$ as

$$\hat{\mu}_N = \frac{1}{N} \sum_{i=1}^{N} \hat{x}_i \quad \text{and} \quad \hat{\Sigma}_N = \frac{1}{N-1} \sum_{i=1}^{N} (\hat{x}_i - \hat{\mu}_N)(\hat{x}_i - \hat{\mu}_N)^\top.$$

We now construct two probability distributionss based on $\hat{\mu}_N$ and $\hat{\Sigma}_N$, that is, we set

$$\hat{\mathbb{P}}_{N,0} = \mathcal{N}(0, \hat{\Sigma}_N) \quad \text{and} \quad \hat{\mathbb{P}}_{N,\Sigma} = \mathcal{N}(\hat{\mu}_N, \Sigma).$$

Notice that $\hat{\mathbb{P}}_{N,0}$ has the same mean as the unknown probability distribution $\mathbb{Q}$ that generates the data, while $\hat{\mathbb{P}}_{N,\Sigma}$ has the same covariance matrix as $\mathbb{Q}$. In the following, we define the mean vector estimation error $\delta_{N,\mathrm{mean}}$ and the covariance matrix estimation error $\delta_{N,\mathrm{covariance}}$ as

$$\delta_{N,\mathrm{mean}} = \varphi(\hat{\mathbb{P}}_{N,\Sigma}, \mathbb{Q}) \quad \text{and} \quad \delta_{N,\mathrm{cov}} = \varphi(\hat{\mathbb{P}}_{N,0}, \mathbb{Q}),$$

respectively, where $\varphi(\cdot, \cdot)$ is a dissimilarity measure for distributions that can be set either to the Fisher-Rao metric (see Section 2) or to the Kullback-Leibler divergence (see Section 3).

Figure A.1 shows the average estimation error for different sample sizes $N$, where the average is taken over 500 independent simulation runs. We observe that the error in estimating the covariance matrix is one order of magnitude higher than the error in estimating the mean vector under both the KL divergence and the FR metric.

(a) KL divergence

(b) FR metric

Figure A.1: Average estimation error for different sample sizes $N$ using the KL divergence or the FR metric as a dissimilarity measure.

# B   Optimistic Likelihood Estimation with Ambiguous Mean Vector

We now consider the FR and KL ambiguity sets for the family of Gaussian distributions with a fixed covariance matrix $\hat{\Sigma} \in \mathbb{S}_{++}^n$. We thus consider the manifold $\Theta = \mathbb{R}^n$ of the mean vector $\theta = \mu$. The FR distance induced by the FR metric on this manifold is denoted by $\bar{d}(\cdot, \cdot)$ and is again available in closed form.

**Proposition B.1** (FR distance between Gaussian distributions [2]). *If $\mathcal{N}(\mu_0, \hat{\Sigma})$ and $\mathcal{N}(\mu_1, \hat{\Sigma})$ are Gaussian distributions with identical covariance matrix $\hat{\Sigma} \in \mathbb{S}_{++}^n$ and mean vectors $\mu_0, \mu_1 \in \mathbb{R}^n$, we have*

$$\bar{d}(\mu_0, \mu_1) = \sqrt{(\mu_0 - \mu_1)^\top \hat{\Sigma}^{-1} (\mu_0 - \mu_1)}.$$

Similarly, the KL divergence between two distributions with the same covariance matrix admits a simple analytical expression.

**Proposition B.2** (KL divergence between Gaussian distributions). *For any $\hat{\Sigma} \in \mathbb{S}_{++}^n$ and $\mu_0, \mu_1 \in \mathbb{R}^n$, the KL divergence from $\mathbb{P}_0 = \mathcal{N}(\mu_0, \hat{\Sigma})$ to $\mathbb{P}_1 = \mathcal{N}(\mu_1, \hat{\Sigma})$ amounts to*

$$\mathrm{KL}(\mathbb{P}_0 \parallel \mathbb{P}_1) = \frac{1}{2}(\mu_0 - \mu_1)^\top \hat{\Sigma}^{-1}(\mu_0 - \mu_1).$$

Throughout this section, we denote by $\hat{\mathbb{P}} = \mathcal{N}(\hat{\mu}, \hat{\Sigma})$ and $\mathbb{P} = (\mu, \hat{\Sigma})$ two Gaussian distributions with the same covariance matrix $\hat{\Sigma} \in \mathbb{S}_{++}^n$ but different mean vectors $\hat{\mu}, \mu \in \mathbb{R}^n$, respectively. Propositions B.1 and B.2 imply that the FR distance and the KL divergence of $\hat{\mathbb{P}}$ and $\mathbb{P}$ satisfy the relation[1] $2 \mathrm{KL}(\hat{\mathbb{P}} \parallel \mathbb{P}) = \bar{d}^2(\hat{\mu}, \mu)$.

With the Fisher-Rao distance as our dissimilarity measure $\varphi$ and given the observations $x_1^M$, the optimistic likelihood problem (2) becomes

$$\min_\mu \left\{ \frac{1}{M} \sum_{m=1}^M (x_m - \mu)^\top \hat{\Sigma}^{-1}(x_m - \mu) + \log \det \hat{\Sigma} : \ (\mu - \hat{\mu})^\top \hat{\Sigma}^{-1}(\mu - \hat{\mu}) \leq \rho^2 \right\}. \quad (A.1)$$

Problem (A.1) is already a finite convex program but can be further simplified to a univariate convex optimization problem and therefore solved in quasi-closed form.

**Theorem B.3** (Optimistic likelihood with mean ambiguity set). *For any $\hat{\mu} \in \mathbb{R}^n, \hat{\Sigma} \in \mathbb{S}_{++}^n$ and $\rho > 0$, the optimal value of problem (A.1) is given by*

$$\frac{1}{M} \sum_{m=1}^M (x_m - \mu^\star)^\top \hat{\Sigma}^{-1}(x_m - \mu^\star) + \log \det \hat{\Sigma},$$

*where $\mu^\star = (1 + \gamma^\star)^{-1} (\bar{x} + \gamma^\star \hat{\mu})$, and $\gamma^\star$ solves the univariate convex optimization problem*

$$\min_{\gamma \geq 0} \ \gamma \left( \rho^2 - \hat{\mu}^\top \hat{\Sigma}^{-1} \hat{\mu} \right) + \frac{(\bar{x} + \gamma \hat{\mu})^\top \hat{\Sigma}^{-1} (\bar{x} + \gamma \hat{\mu})}{1 + \gamma} \quad (A.2)$$

*with $\bar{x} = M^{-1} \sum_{m=1}^M x_m$.*

*Proof.* As $\hat{\Sigma}$ is constant, the minimizers of (A.1) also solve

$$\min_\mu \left\{ M^{-1} \sum_{m=1}^M (x_m - \mu)^\top \hat{\Sigma}^{-1}(x_m - \mu) : \ (\mu - \hat{\mu})^\top \hat{\Sigma}^{-1}(\mu - \hat{\mu}) \leq \rho^2 \right\}. \quad (A.3)$$

Problem (A.3) is equivalent to

$$\min_\mu \max_{\gamma \geq 0} \left\{ \ \left\langle \hat{\Sigma}^{-1}, M^{-1} \sum_{m=1}^M (\mu - x_m)(\mu - x_m)^\top \right\rangle + \gamma \left( \left\langle \hat{\Sigma}^{-1}, (\mu - \hat{\mu})(\mu - \hat{\mu})^\top \right\rangle - \rho^2 \right) \ \right\}$$

$$= \max_{\gamma \geq 0} \min_\mu \left\{ \ -\gamma \rho^2 + \left\langle \hat{\Sigma}^{-1}, M^{-1} \sum_{m=1}^M (\mu - x_m)(\mu - x_m)^\top + \gamma (\mu - \hat{\mu})(\mu - \hat{\mu})^\top \right\rangle \ \right\},$$

where the equality follows from strong duality, which holds because $\rho > 0$ and because $\mu = \hat{\mu}$ constitutes a Slater point for the primal problem (A.3). For any fixed $\gamma \geq 0$, the inner minimization problem over $\mu$ admits the optimal solution

$$\mu^\star(\gamma) = (1 + \gamma)^{-1} (\bar{x} + \gamma \hat{\mu})$$

with $\bar{x} = M^{-1} \sum_{m=1}^{M} x_m$. Thus, the optimal value of (A.3) equals

$$\max_{\gamma \geq 0} \gamma \left( \hat{\mu}^\top \hat{\Sigma}^{-1} \hat{\mu} - \rho^2 \right) - \frac{(\bar{x} + \gamma \hat{\mu})^\top \hat{\Sigma}^{-1} (\bar{x} + \gamma \hat{\mu})}{1 + \gamma},$$

which is equivalent to the minimization problem (A.2). By strong duality, given any minimizer $\gamma^\star$ of problem (A.2), an optimal solution for (A.3) and also for (A.1) can be constructed as

$$\mu^\star = (1 + \gamma^\star)^{-1} \left( \bar{x} + \gamma^\star \hat{\mu} \right).$$

Substituting $\mu^\star$ into the objective function of (A.1) yields the postulated optimal value. $\square$

In the following, we provide the first- and second-order derivatives of the objective function of (A.2), which can be used for implementing the optimization algorithm to solve for $\gamma^\star$. To this end, we denote by $g(\gamma)$ the objective function of (A.2). A direct calculation shows that

$$g'(\gamma) = \left( \rho^2 - \hat{\mu}^\top \hat{\Sigma}^{-1} \hat{\mu} \right) + \frac{((2 + \gamma)\hat{\mu} - \bar{x})^\top \hat{\Sigma}^{-1} (\bar{x} + \gamma \hat{\mu})}{(1 + \gamma)^2}.$$

Moreover, the second-order derivative of $g(\gamma)$ is given by

$$g''(\gamma) = \frac{2 \hat{\mu}^\top \hat{\Sigma}^{-1} \hat{\mu}}{(1 + \gamma)} - \frac{2[(2 + \gamma)\hat{\mu} - \bar{x}]^\top \hat{\Sigma}^{-1} (\bar{x} + \gamma \hat{\mu})}{(1 + \gamma)^3}.$$

## C  Proofs of Section 2

To prove Lemma 2.2, we require the following preparatory lemma.

**Lemma C.1** (Properties of $\mathcal{B}^{\mathrm{FR}}$)**.**  The FR ball has the following properties:

(i)   $\mathcal{B}^{\mathrm{FR}}$ is compact and complete on $\mathbb{S}_{++}^n$.
(ii)  For any $\Sigma \in \mathcal{B}^{\mathrm{FR}}$, we have $\lambda_{\min}(\hat{\Sigma}) e^{-\sqrt{2}\rho} \cdot I_n \preceq \Sigma \preceq \lambda_{\max}(\hat{\Sigma}) e^{\sqrt{2}\rho} \cdot I_n$.

*Proof.* To prove assertion (i), we first show that $\mathcal{B}^{\mathrm{FR}}$ is compact and complete with respect to the topology induced by the Riemannian distance $d(\cdot, \cdot)$. Recall that $\mathbb{S}_{++}^n$ is a Hadamard manifold and thus constitutes a complete metric space. By the Hopf-Rinow theorem [6, § 8, Theorem 2.8(b)], $\mathcal{B}^{\mathrm{FR}}$ is compact in the usual topology because $\mathcal{B}^{\mathrm{FR}}$ is a metric ball and therefore closed and bounded. Moreover, $\mathcal{B}^{\mathrm{FR}}$ is complete in the usual topology because any closed subset of a complete metric space is complete as well. By [10, Theorem 13.29], the metric topology with respect to $d(\cdot, \cdot)$ on $\mathbb{S}_{++}^n$ coincides with the subspace topology of $\mathbb{S}_{++}^n$ with respect to the usual topology on $\mathbb{S}^n$. This completes the proof of assertion (i).

To prove assertion (ii), pick any $\Sigma \in \mathcal{B}^{\mathrm{FR}}$ and let $0 \leq \lambda_1(A) \leq \ldots \leq \lambda_n(A)$ denote the eigenvalues of any symmetric positive definite $n$-by-$n$ matrix $A$ in increasing order. Then, we have

$$\sqrt{\log^2(\lambda_i(\hat{\Sigma}^{-\frac{1}{2}} \Sigma \hat{\Sigma}^{-\frac{1}{2}}))} \leq \sqrt{\sum_{j=1}^{n} \log^2(\lambda_j(\hat{\Sigma}^{-\frac{1}{2}} \Sigma \hat{\Sigma}^{-\frac{1}{2}}))} = \| \log(\hat{\Sigma}^{-\frac{1}{2}} \Sigma \hat{\Sigma}^{-\frac{1}{2}}) \|_F \leq \sqrt{2}\rho$$

for any $i = 1, \ldots, n$, where the equality follows from the definition of the Frobenius norm, and the last inequality follows from the definition of $\mathcal{B}^{\mathrm{FR}}$. Note that $\lambda_i(\hat{\Sigma}^{-\frac{1}{2}} \Sigma \hat{\Sigma}^{-\frac{1}{2}}) = 1/\lambda_{n-i+1}(\Sigma^{-\frac{1}{2}} \hat{\Sigma} \Sigma^{-\frac{1}{2}})$, and hence any eigenvalue $\lambda_i(\hat{\Sigma}^{-\frac{1}{2}} \Sigma \hat{\Sigma}^{-\frac{1}{2}})$ obeys the bounds

$$e^{-\sqrt{2}\rho} \leq \lambda_i(\hat{\Sigma}^{-\frac{1}{2}} \Sigma \hat{\Sigma}^{-\frac{1}{2}}) \leq e^{\sqrt{2}\rho}.$$

This implies that

$$\lambda_{\max}^{-1}(\hat{\Sigma}) \lambda_{\max}(\Sigma) \leq e^{\sqrt{2}\rho} \quad \text{and} \quad \lambda_{\min}^{-1}(\hat{\Sigma}) \lambda_{\min}(\Sigma) \geq e^{-\sqrt{2}\rho},$$

which completes the proof of assertion (ii). $\square$

We are now ready to prove Lemma 2.2.

*Proof of Lemma 2.2.* First, assertion (i) of Lemma C.1 ensures that the feasible region $\mathcal{B}^{\mathrm{FR}}$ is compact. Second, we note that the objective function $L(\cdot)$ is continuous at any positive definite matrix. By assertion (ii) of Lemma C.1, there is a uniform positive lower bound on the eigenvalues of all matrices in $\mathcal{B}^{\mathrm{FR}}$. Therefore $L(\cdot)$ is continuous on $\mathcal{B}^{\mathrm{FR}}$. The solvability of problem (6) then follows from Weierstrass' extreme value theorem [1, Corollary 2.35]. $\qquad\square$

*Proof of Theorem 2.5.* We first show that $\mathcal{B}^{\mathrm{FR}}$ is a geodesically convex set. By [5, Proposition II.1.4], balls in CAT($\kappa$) spaces[2] of radius less than $D_\kappa/2$ are geodesically convex, where $D_\kappa$ is the diameter of the model space of constant curvature $\kappa$ (see [5, Definition I.2.10]). It is known that the smooth manifold $\mathbb{S}_{++}^n$ is a CAT(0) space [5, Theorem II.10.39], which implies via [5, Point I.2.12] that $D_0 = \infty$. The claim thus follows.

The proof that $L(\cdot)$ is a geodesically convex function over $\mathbb{S}_{++}^n$ closely follows from [15, Lemma III.2] and [12, Corollary 5.3] and is thus omitted. $\qquad\square$

*Proof of Lemma 2.6.* The claim trivially holds if $\rho' \leq \rho$. We thus prove the two statements under the assumption that $\rho' > \rho$. By [5, Theorem II.10.39], $\mathbb{S}_{++}^n$ is a CAT(0) space. Furthermore, by Lemma C.1, the geodesic ball $\mathcal{B}^{\mathrm{FR}}$ is both complete and compact. Assertion (i) in Lemma 2.6 then follows from [5, Proposition II.2.4].

To prove assertion (ii), we define

$$\Sigma_p \triangleq \hat{\Sigma}^{\frac{1}{2}} (\hat{\Sigma}^{-\frac{1}{2}} \Sigma' \hat{\Sigma}^{-\frac{1}{2}})^{\frac{\rho}{\rho'}} \hat{\Sigma}^{\frac{1}{2}}.$$

One readily verifies that $d(\hat{\Sigma}, \Sigma_p) = \rho$, and hence $\Sigma_p \in \mathcal{B}^{\mathrm{FR}}$. Recall that $\Sigma' \in \mathbb{S}_{++}^n$ and $d(\hat{\Sigma}, \Sigma') = \rho'$. Given any $\Sigma \in \mathcal{B}^{\mathrm{FR}}$, by the triangle inequality, we thus have

$$d(\Sigma, \Sigma') \geq d(\hat{\Sigma}, \Sigma') - d(\hat{\Sigma}, \Sigma) \geq d(\hat{\Sigma}, \Sigma') - \max_{\Sigma'' \in \mathcal{B}^{\mathrm{FR}}} d(\hat{\Sigma}, \Sigma'') = \rho' - \rho.$$

This reasoning implies that

$$\min_{\Sigma \in \mathcal{B}^{\mathrm{FR}}} d(\Sigma, \Sigma') \geq \rho' - \rho. \tag{A.4}$$

By definition, the geodesic $\gamma(t) = \hat{\Sigma}^{\frac{1}{2}} (\hat{\Sigma}^{-\frac{1}{2}} \Sigma' \hat{\Sigma}^{-\frac{1}{2}})^t \hat{\Sigma}^{\frac{1}{2}}$ connecting $\hat{\Sigma}$ and $\Sigma'$ has constant-speed, that is, $d(\gamma(t), \gamma(s)) = d(\gamma(0), \gamma(1)) \cdot |t - s|$ for any $t, s \in [0, 1]$ (see [4, Theorem 6.1.6]). Therefore, we have

$$d(\Sigma_p, \Sigma') = d(\gamma(\tfrac{\rho}{\rho'}), \gamma(1)) = d(\hat{\Sigma}, \Sigma') \cdot \left| \frac{\rho}{\rho'} - 1 \right| = \rho' - \rho,$$

which implies that the lower bound (A.4) is attained by $\Sigma_p$. The uniqueness result of assertion (i) thus allows us to conclude that $\Sigma_p$ is the projection of $\Sigma'$ onto $\mathcal{B}^{\mathrm{FR}}$. $\qquad\square$

The proof of Theorem 2.7 is based on the following two technical lemmas.

**Lemma C.2** (Bounded gradient). *For any $X \in T_\Sigma \mathbb{S}_{++}^n$, denote by $\|X\|_\Sigma \triangleq \sqrt{\langle X, X \rangle_\Sigma}$ the norm induced by the inner product $\langle \cdot, \cdot \rangle_\Sigma$ defined in (8). The Riemannian gradient of the objective function $L(\cdot)$ of problem (6) satisfies*

$$\| \mathrm{grad}\, L(\Sigma) \|_\Sigma \leq \sqrt{n} \cdot e^{2\sqrt{2}\rho} \cdot \lambda_{\min}^{-2}(\hat{\Sigma}) \cdot \max\{|1 - e^{\sqrt{2}\rho} \lambda_{\min}^{-1}(\hat{\Sigma}) \lambda_{\max}(S)|, 1\} \quad \forall \Sigma \in \mathcal{B}^{\mathrm{FR}}.$$

*Proof.* By (9) and the definition of $\| \cdot \|_\Sigma$, we have

$$\| \mathrm{grad} L(\Sigma) \|_\Sigma^2 = \frac{1}{2} \mathrm{Tr} \left( \mathrm{grad} L(\Sigma) \cdot \Sigma^{-1} \cdot \mathrm{grad} L(\Sigma) \cdot \Sigma^{-1} \right) = \frac{1}{2} \mathrm{Tr} \left( A\Sigma^{-2} A \Sigma^{-2} \right),$$

where $A \triangleq (I_n - \Sigma^{-\frac{1}{2}} S \Sigma^{-\frac{1}{2}})$. Lemma C.1(ii) thus implies that

$$(1 - e^{\sqrt{2}\rho} \lambda_{\min}^{-1}(\hat{\Sigma}) \lambda_{\max}(S)) I_n \preceq (1 - \lambda_{\min}^{-1}(\Sigma) \lambda_{\max}(S)) I_n \preceq A \preceq I_n,$$

and therefore we have

$$\|\mathrm{grad}L(\Sigma)\|_\Sigma \leq \sqrt{\frac{n}{2} \cdot \frac{\lambda^2_{\max}(A)}{\lambda^4_{\min}(\Sigma)}}$$

$$\leq \sqrt{\frac{n}{2} \cdot \frac{\max\{1, (1 - e^{\sqrt{2}\rho}\lambda^{-1}_{\min}(\hat{\Sigma})\lambda_{\max}(S))^2\}}{\lambda^4_{\min}(\hat{\Sigma})e^{-4\sqrt{2}\rho}}}$$

$$= \frac{\sqrt{n} \cdot \max\left\{1, \left|1 - e^{\sqrt{2}\rho}\lambda^{-1}_{\min}(\hat{\Sigma})\lambda_{\max}(S)\right|\right\}}{\sqrt{2}\lambda^2_{\min}(\hat{\Sigma})e^{-2\sqrt{2}\rho}},$$

where the last inequality follows from Lemma C.1(ii). This observation completes the proof. $\square$

**Lemma C.3** (Lower bounded sectional curvature). *The sectional curvature of the Riemannian manifold $\mathbb{S}^n_{++}$ equipped with the FR metric (8) is lower bounded by $-2$.*

*Proof.* Select $\Sigma \in \mathbb{S}^n_{++}$, and let $X, Y \in T_\Sigma \mathbb{S}^n_{++}$ be two orthonormal tangent vectors at $\Sigma$, that is,

$$\|X\|_\Sigma = 1 = \|Y\|_\Sigma \quad \text{and} \quad \langle X, Y \rangle_\Sigma = 0.$$

Using the formula for the Riemannian curvature tensor $R(\cdot, \cdot, \cdot, \cdot)$ from [11, Theorem 2.1 (ii)], we have

$$R(X, Y, Y, X) = -\frac{1}{4}\mathrm{Tr}\left(Y\Sigma^{-1}X\Sigma^{-1}X\Sigma^{-1}Y\Sigma^{-1}\right) + \frac{1}{4}\mathrm{Tr}\left(X\Sigma^{-1}Y\Sigma^{-1}X\Sigma^{-1}Y\Sigma^{-1}\right). \quad \text{(A.5)}$$

Then, the sectional curvature $\kappa(X, Y)$ associated with the 2-plane spanned by $\{X, Y\}$ satisfies

$$\kappa(X, Y) = - R(X, Y, X, Y)$$
$$= -\frac{1}{4}\mathrm{Tr}\left(Y\Sigma^{-1}X\Sigma^{-1}X\Sigma^{-1}Y\Sigma^{-1}\right) + \frac{1}{4}\mathrm{Tr}\left(X\Sigma^{-1}Y\Sigma^{-1}X\Sigma^{-1}Y\Sigma^{-1}\right)$$
$$\geq - \left(\|X\|^2_\Sigma\|Y\|^2_\Sigma + \|X\|^2_\Sigma\|Y\|^2_\Sigma\right) = -2,$$

where the first equality follows from [9, Proposition 8.8], the second equality exploits (A.5), and the inequality holds due to the Cauchy-Schwarz inequality. $\square$

We are now equipped with all the necessary ingredients to prove Theorem 2.7.

*Proof of Theorem 2.7.* The proof closely follows that of [14, Theorem 9]. The main difference is that we replace the assumption of Lipschitz continuity of the objective function with the assumption of a bounded Riemannian gradient. Due to Theorem 2.5, the function $L(\cdot)$ is geodesically convex. So we have (see the sentence following Definition 2 in [14])

$$L(\Sigma') \geq L(\Sigma) + \langle \mathrm{grad}L(\Sigma), \mathrm{Exp}^{-1}_\Sigma(\Sigma')\rangle_\Sigma, \quad \forall \Sigma, \Sigma' \in \mathbb{S}^n_{++}.$$

Therefore, for any $k \geq 1$,

$$L(\Sigma_k) - L(\Sigma^\star) \leq -\langle \mathrm{grad}L(\Sigma_k), \mathrm{Exp}^{-1}_{\Sigma_k}(\Sigma^\star)\rangle_{\Sigma_k}. \quad \text{(A.6)}$$

By [14, Corollary 8], Lemma C.3 and because the diameter of the feasible region is $2\rho$, the right hand side of (A.6) is upper bounded by

$$\frac{1}{2\alpha}\left(d^2(\Sigma_k, \Sigma^\star) - d^2(\Sigma_{k+1}, \Sigma^\star)\right) + \frac{\alpha \cdot (2\rho) \cdot \sqrt{2} \cdot \|\mathrm{grad}L(\Sigma_k)\|^2_{\Sigma_k}}{2\tanh((2\rho) \cdot \sqrt{2})}, \quad \text{(A.7)}$$

where the norm $\|\cdot\|_{\Sigma_k}$ is defined as in Lemma C.2. Therefore, substituting the upper bound (A.7) into (A.6) and using C.2, we find

$$L(\Sigma_k) - L(\Sigma^\star) \leq \frac{1}{2\alpha}\left(d^2(\Sigma_k, \Sigma^\star) - d^2(\Sigma_{k+1}, \Sigma^\star)\right) + \frac{\sqrt{2}\alpha\rho\Gamma^2}{\tanh(2\sqrt{2}\rho)}, \quad \text{(A.8)}$$

where $\Gamma \triangleq 2^{-1/2}\sqrt{n} \cdot e^{2\sqrt{2}\rho} \cdot \lambda_{\min}^{-2}(\hat{\Sigma}) \cdot \max\{|1 - e^{\sqrt{2}\rho}\lambda_{\min}^{-1}(\hat{\Sigma})\lambda_{\max}(S)|, 1\}$. By telescoping, we then obtain

$$\frac{1}{K}\sum_{k=1}^{K} L(\Sigma_k) - L(\Sigma^\star) \leq \frac{1}{2\alpha K}\left(d^2(\Sigma_1, \Sigma^\star) - d^2(\Sigma_{K+1}, \Sigma^\star)\right) + \frac{\sqrt{2}\alpha\rho\Gamma^2}{\tanh(2\sqrt{2}\rho)}$$

$$\leq \frac{2\rho^2}{\alpha K} + \frac{\sqrt{2}\alpha\rho\Gamma^2}{\tanh(2\sqrt{2}\rho)} = \frac{2^{\frac{7}{4}}\rho^{\frac{3}{2}}\Gamma}{\sqrt{K\tanh(2\sqrt{2}\rho)}},$$

where the second inequality follows from the bounds $d^2(\Sigma_{K+1}, \Sigma^\star) \geq 0$ and $d(\Sigma_1, \Sigma^\star) \leq 2\rho$, and the equality holds because $\alpha = 2^{1/4}\sqrt{\rho\tanh(2\sqrt{2}\rho)}/(\Gamma\sqrt{K})$. Note that although the matrix $\Sigma_{K+1}$ is not actually computed because Algorithm 1 is terminated at $k = K - 1$, it is well-defined, and inequality (A.8) is valid for $k = K$. If we can show that

$$L(\bar{\Sigma}_K) \leq \frac{1}{K}\sum_{k=1}^{K} L(\Sigma_k),$$

the desired result follows. Towards that end, we prove by induction that

$$L(\bar{\Sigma}_T) \leq \frac{1}{T}\sum_{k=1}^{T} L(\Sigma_k) \quad \forall T \in \mathbb{N}t. \tag{A.9}$$

The inequality trivially holds for $T = 1$. Suppose now that the inequality in (A.9) holds for some $T \geq 1$. Then, we have

$$L(\bar{\Sigma}_{T+1}) = L\left(\bar{\Sigma}_T^{\frac{1}{2}}\left(\bar{\Sigma}_T^{-\frac{1}{2}}\Sigma_{T+1}\bar{\Sigma}_T^{-\frac{1}{2}}\right)^{\frac{1}{T+1}}\bar{\Sigma}_T^{\frac{1}{2}}\right)$$

$$= L\left(\gamma_T\left(\frac{1}{T+1}\right)\right)$$

$$\leq \frac{T}{T+1}L\left(\bar{\Sigma}_T\right) + \frac{1}{T+1}L\left(\Sigma_{T+1}\right)$$

$$\leq \frac{1}{T+1}\sum_{k=1}^{T} L(\Sigma_k) + \frac{1}{T+1}L\left(\Sigma_{T+1}\right)$$

$$= \frac{1}{T+1}\sum_{k=1}^{T+1} L(\Sigma_k),$$

where $\gamma_T$ denotes the geodesic from $\bar{\Sigma}_T$ to $\Sigma_{T+1}$. The first inequality follows from the geodesic convexity of $L(\cdot)$ (see Theorem 2.5 and Definition 2.4), and the second inequality holds due to the induction hypothesis (A.9). The claim now follows because $T$ was chosen arbitrarily. $\square$

Next, we formally define the notions of geodesic strong convexity and geodesic smoothness for functions on $\mathbb{S}_{++}^n$.

**Definition C.4** (Strong convexity). Let $\mathcal{B} \subseteq \mathbb{S}_{++}^n$ be a subset and $\sigma > 0$. A differentiable function $F : \mathcal{B} \to \mathbb{R}$ is said to be (geodesically) $\sigma$-strongly convex on $\mathcal{B}$ if

$$F(Y) \geq F(X) + \langle \operatorname{grad} F(X), \operatorname{Exp}_X^{-1}(Y)\rangle_X + \frac{\sigma}{2}d^2(X, Y). \tag{A.10}$$

**Definition C.5** (Smoothness). Let $\mathcal{B} \subseteq \mathbb{S}_{++}^n$ be a subset and $\beta > 0$. A differentiable function $F : \mathcal{B} \to \mathbb{R}$ is said to be (geodesically) $\beta$-smooth on $\mathcal{B}$ if

$$F(Y) \leq F(X) + \langle \operatorname{grad} F(X), \operatorname{Exp}_X^{-1}(Y)\rangle_X + \frac{\beta}{2}d^2(X, Y). \tag{A.11}$$

The proof of Lemma 2.8 is based on the following preparatory results.

**Lemma C.6.** Let $F : \mathbb{S}^n_{++} \to \mathbb{R}$ be a twice continuously differentiable function and $\mathcal{B} \subseteq \mathbb{S}^n_{++}$ be a geodesically convex subset. The following implications hold.

(i) If the smallest eigenvalue of the Riemannian Hessian $\operatorname{hess} F(X)$ (interpreted as an operator on $T_X \mathbb{S}^N_{++}$) of $F$ at $X$ is lower bounded uniformly on $\mathcal{B}$ by $\sigma > 0$, *i.e.*,

$$\min \left\{ \langle \operatorname{hess} F(X)[V], V \rangle : V \in T_X \mathbb{S}^N_{++}, \|V\|_X = 1 \right\} \geq \sigma \quad \forall X \in \mathcal{B}, \tag{A.12}$$

then $F$ is $\sigma$-strongly convex on $\mathcal{B}$.

(ii) If the largest eigenvalue of the Riemannian Hessian $\operatorname{hess} F(X)$ (interpreted as an operator on $T_X \mathbb{S}^N_{++}$) of $F$ at $X$ is upper bounded uniformly on $\mathcal{B}$ by $\beta > 0$, *i.e.*,

$$\max \left\{ \langle \operatorname{hess} F(X)[V], V \rangle : V \in T_X \mathbb{S}^N_{++}, \|V\|_X = 1 \right\} \leq \beta \quad \forall X \in \mathcal{B}, \tag{A.13}$$

then $F$ is $\beta$-smooth on $\mathcal{B}$.

The proof of Lemma C.6 closely follows that of its Euclidean counterpart and is omitted here.

*Proof of Lemma 2.8.* Define $f(\Sigma) = \operatorname{Tr}\left(\Sigma^{-1} S\right)$. Because $\log \det \Sigma$ is a geodesically linear function [13, Proposition 12], it suffices to study the smoothness and convexity properties of $f(\cdot)$. By [7, Equations (28)], the Riemannian Hessian $\operatorname{hess} f(\Sigma)$ at $\Sigma$ is given by

$$\operatorname{hess} f(\Sigma)[V] = \Sigma \left(\nabla^2 f(\Sigma)[V]\right) \Sigma + \frac{1}{2} \left(V \nabla f(\Sigma) \Sigma + \Sigma \nabla f(\Sigma) V\right) \quad \forall V \in T_\Sigma \mathbb{S}^n_{++}. \tag{A.14}$$

By elementary matrix calculus, we know that

$$\nabla f(\Sigma) = -\Sigma^{-1} S \Sigma^{-1} \tag{A.15}$$

and

$$\nabla^2 f(\Sigma)[V] = \Sigma^{-1} V \Sigma^{-1} S \Sigma^{-1} + \Sigma^{-1} S \Sigma^{-1} V \Sigma^{-1} \quad \forall V \in \mathbb{S}^n, \tag{A.16}$$

where the Hessian $\nabla^2 f(\Sigma)$ is interpreted as a linear operator on $\mathbb{S}^n$. Noting that $T_\Sigma \mathbb{S}^n_{++} = \mathbb{S}^n$ and combining (A.14), (A.15) and (A.16), we obtain

$$\langle \operatorname{hess} f(\Sigma)[V], V \rangle_\Sigma = \operatorname{Tr}\left(\Sigma^{-1} S \Sigma^{-1} V \Sigma^{-1} V\right) \quad \forall V \in \mathbb{S}^n. \tag{A.17}$$

Using these preparatory results, we now demonstrate that $f(\cdot)$ is $\beta$-smooth and $\sigma$-strongly convex in the geodesic sense.

**Smoothness.** In order to establish the smoothness properties of $f(\cdot)$, we consider the maximization problem

$$\max \left\{ \langle \operatorname{hess} f(\Sigma)[V], V \rangle_\Sigma : V \in \mathbb{S}^n, \|V\|_\Sigma = 1 \right\},$$

which, by (A.17) and the definition of $\|\cdot\|_\Sigma$, is equivalent to

$$\max \left\{ \operatorname{Tr}\left(\Sigma^{-1} S \Sigma^{-1} V \Sigma^{-1} V\right) : V \in \mathbb{S}^n, \tfrac{1}{2} \operatorname{Tr}\left(\Sigma^{-1} V \Sigma^{-1} V\right) = 1 \right\}.$$

The optimal value of this problem is upper bounded by $2\,\lambda_{\max}(S)/\lambda_{\min}(\Sigma)$. Using the bound from Lemma C.1(ii), we have

$$\frac{2\,\lambda_{\max}(S)}{\lambda_{\min}(\Sigma)} \leq \frac{2\lambda_{\max}(S)}{\lambda_{\min}(\widehat{\Sigma}) \exp(-\sqrt{2}\rho)} = \beta.$$

By Lemma C.6(ii), $f(\cdot)$ is $\beta$-smooth.

**Strong convexity.** In order to establish the convexity properties of $f(\cdot)$, we consider the minimization problem

$$\min \left\{ \langle \operatorname{hess} f(\Sigma)[V], V \rangle_\Sigma : V \in \mathbb{S}^n, \|V\|_\Sigma = 1 \right\},$$

which, by (A.17) and the definition of $\|\cdot\|_\Sigma$, is equivalent to

$$\min \left\{ \operatorname{Tr}\left(\Sigma^{-1} S \Sigma^{-1} V \Sigma^{-1} V\right) : V \in \mathbb{S}^n, \tfrac{1}{2} \operatorname{Tr}\left(\Sigma^{-1} V \Sigma^{-1} V\right) = 1 \right\}.$$

The optimal value of this problem is lower bounded by $2\,\lambda_{\min}(S)/\lambda_{\max}(\Sigma)$. Using the bound in Lemma C.1(ii), we have

$$\frac{2\,\lambda_{\min}(S)}{\lambda_{\max}(\Sigma)} = \frac{2\lambda_{\min}(S)}{\lambda_{\max}(\widehat{\Sigma}) \exp(\sqrt{2}\rho)} = \sigma.$$

Since $S \succ 0$, $\sigma > 0$. By Lemma C.6(i), $f(\cdot)$ is thus $\sigma$-strongly convex. This completes the proof. $\square$

# D Proofs of Section 3

*Proof of Theorem 3.2.* By applying the change of variables $Z \leftarrow \Sigma^{-1}$, problem (12) can be reformulated as

$$\inf_Z \left\{ \mathrm{Tr}\left(SZ\right) - \log \det Z : Z \succ 0, \ \mathrm{Tr}\left(\hat\Sigma Z\right) - \log \det Z \leq \bar\rho \right\}, \tag{A.18}$$

where $\bar\rho \triangleq 2\rho + n + \log \det \hat\Sigma$. Note that (A.18) is equivalent to

$$\inf_{Z \succ 0} \sup_{\gamma \geq 0} \ \mathrm{Tr}\left(SZ\right) - \log \det Z + \gamma \left( \mathrm{Tr}\left(\hat\Sigma Z\right) - \log \det Z - \bar\rho \right)$$

$$= \sup_{\gamma \geq 0} \inf_{Z \succ 0} \ -\gamma\bar\rho + \mathrm{Tr}\left((S + \gamma\hat\Sigma)Z\right) - (1+\gamma) \log \det Z$$

$$= \sup_{\gamma \geq 0} \left\{ -\gamma\bar\rho + \inf_{Z \succ 0} \left\{ \mathrm{Tr}\left((S + \gamma\hat\Sigma)Z\right) - (1+\gamma) \log \det Z \right\} \right\}, \tag{A.19}$$

where the first equality follows from strong duality, which holds because $\rho > 0$ and because $\hat\Sigma^{-1}$ is a Slater point for the primal problem (A.18).

To analyze problem (A.19), assume first that $S$ is singular. If $\gamma = 0$, then the inner minimization problem over $Z$ is unbounded, and thus $\gamma = 0$ is never optimal for the outer maximization problem. For any $\gamma > 0$, the inner minimization problem over $Z$ admits the optimal solution $Z^\star(\gamma) = (1+\gamma)(S + \gamma\hat\Sigma)^{-1}$. Problem (A.18) is thus equivalent to

$$\sup_{\gamma > 0} \left\{ -\gamma\bar\rho + n(1+\gamma) - (1+\gamma) \log \det[(1+\gamma)(S + \gamma\hat\Sigma)^{-1}] \right\}$$

$$= \sup_{\gamma > 0} \left\{ -\gamma\bar\rho + n(1+\gamma) - (1+\gamma) \log[(1+\gamma)^n \det(S + \gamma\hat\Sigma)^{-1}] \right\}$$

$$= \sup_{\gamma > 0} \left\{ -\gamma\bar\rho + n(1+\gamma) - n(1+\gamma) \log(1+\gamma) - (1+\gamma) \log \det(S + \gamma\hat\Sigma)^{-1} \right\}. \tag{A.20}$$

By strong duality, any minimizer $\gamma^\star$ of (13) can be used to construct a minimizer

$$\Sigma^\star = (1+\gamma^\star)^{-1}(S + \gamma^\star\hat\Sigma)$$

for problem (12). This observation establishes the claim if $S$ is singular.

Assume next that $S$ has full rank. In this case, the inner minimization problem in (A.19) admits the optimal solution $Z^\star(\gamma) = (1+\gamma)(S + \gamma\hat\Sigma)^{-1}$ for any fixed $\gamma \geq 0$, and thus problem (A.19) is equivalent to

$$\sup_{\gamma \geq 0} \left\{ -\gamma\bar\rho + n(1+\gamma) - n(1+\gamma) \log(1+\gamma) - (1+\gamma) \log \det(S + \gamma\hat\Sigma)^{-1} \right\},$$

which differs from (A.20) only in that it has a closed feasible set, that is, $\gamma = 0$ is feasible. Because the objective function of the above optimization problem is continuous in $\gamma$, we can in fact optimize over $\gamma > 0$ without reducing the supremum. The claim now follows by replacing $\bar\rho$ with its definition and eliminating the constant term from the objective function. $\square$

*Proof of Corollary 3.3.* For any $\hat\Sigma \in \mathbb{S}^n_{++}$ and $\gamma > 0$, the Woodbury formula [3, Corollary 2.8.8] implies that

$$(S + \gamma\hat\Sigma)^{-1} = \gamma^{-1}\hat\Sigma^{-\frac{1}{2}}(\gamma^{-1}\hat\Sigma^{-\frac{1}{2}} S\hat\Sigma^{-\frac{1}{2}} + I_n)^{-1}\hat\Sigma^{-\frac{1}{2}},$$

and thus we have

$$\log \det(\gamma\hat\Sigma + S)^{-1} + n \log \gamma + \log \det \hat\Sigma = -\log \det \left( I_n + \gamma^{-1}\hat\Sigma^{-\frac{1}{2}} \Lambda\Lambda^\top \hat\Sigma^{-\frac{1}{2}} \right)$$

$$= -\log \det \left( I_k + \gamma^{-1}\Lambda^\top \hat\Sigma^{-1}\Lambda \right)$$

$$= k \log \gamma - \log \det \left( \gamma I_k + \Lambda^\top \hat\Sigma^{-1}\Lambda \right),$$

where the second equality follows from [3, Equation 2.8.14]. Substituting the expression for $\log \det(\gamma\hat{\Sigma} + S)^{-1}$ into (13) and removing the irrelevant constant term $(n + \log \det \hat{\Sigma})$ yields the equivalent minimization problem

$$\inf_{\gamma>0} \left\{ 2\gamma\rho + n(1+\gamma)\log(1+\gamma) - (n-k)(1+\gamma)\log\gamma - (1+\gamma)\log\det(\gamma I_k + \Lambda^\top\hat{\Sigma}^{-1}\Lambda) \right\}.$$

This observation completes the proof. □

## E  Derivatives of Problem (13)

Use $g_1(\gamma)$ as a shorthand for the objective function of problem (13). In the following, we provide the first- and second-order derivatives of $g_1(\cdot)$, which are needed by the optimization algorithm that solves (13). In particular, the first-order derivative is given by

$$g_1'(\gamma) = 2\rho + n\left(\log(1+\gamma) + 1\right) - \log\det\left(\gamma I_n + S\hat{\Sigma}^{-1}\right) - (1+\gamma)\operatorname{Tr}\left((\gamma I_n + S\hat{\Sigma}^{-1})^{-1}\right),$$

and the second-order derivative can be expressed as

$$g_1''(\gamma) = \frac{n}{1+\gamma} - \operatorname{Tr}\left((\gamma I_n + S\hat{\Sigma}^{-1})^{-1}\left(2I_n + (1+\gamma)(\gamma I_n + S\hat{\Sigma}^{-1})^{-1}\right)\right).$$

Next, denote by $g_2$ the objective function of the singular reduction problem of Corollary 3.3, that is,

$$g_2(\gamma) = 2\gamma\rho + n(1+\gamma)\log(1+\gamma) - (n-k)(1+\gamma)\log\gamma - (1+\gamma)\log\det(\gamma I_k + \Lambda^\top\hat{\Sigma}^{-1}\Lambda).$$

The first- and second-order derivative of $g_2$ are given by

$$g_2'(\gamma) = 2\rho + n\left(\log(1+\gamma) + 1\right) - (n-k)\left(\log\gamma + \gamma^{-1} + 1\right)$$
$$- \log\det\left(\gamma I_k + \Lambda^\top\hat{\Sigma}^{-1}\Lambda\right) - (1+\gamma)\operatorname{Tr}\left((\gamma I_k + \Lambda^\top\hat{\Sigma}^{-1}\Lambda)^{-1}\right),$$

and

$$g_2''(\gamma) = \frac{n}{1+\gamma} - (n-k)(\gamma^{-1} - \gamma^{-2})$$
$$- \operatorname{Tr}\left((\gamma I_k + \Lambda^\top\hat{\Sigma}^{-1}\Lambda)^{-1}\left(2I_k + (1+\gamma)(\gamma I_k + \Lambda^\top\hat{\Sigma}^{-1}\Lambda)^{-1}\right)\right),$$

respectively.

## Footnotes

[1] More generally, for arbitrary parametric families of distributions, the second-order Taylor expansion of the KL divergence is given by the FR distance because $\mathrm{KL}(\hat{\mathbb{P}} \parallel \mathbb{P}) = \frac{1}{2} \mathrm{FR}^2(\hat{\mathbb{P}}, \mathbb{P}) + \mathcal{O}(\mathrm{FR}^3(\hat{\mathbb{P}}, \mathbb{P}))$. See [8, § 7.2.2] for further details.

[2]A formal definition of CAT spaces can be found in [5, Definition II.1.1], and the upper bound $\kappa$ of the curvature of a metric space is defined in [5, Definition II.1.2]