[Reviews · NeurIPS 2019]

Reviewer 1



The paper is nicely written and easy to follow. For the modeling proposed in Section 2, it is easy to work with the regularized problem formulation, that is, min_Sigma + log det (Sigma) + lambda d^2(Sigma, \hat{Sigma}) for some lambda ambiguity radius. Consequently, the convexity proof can be made simpler by computing the Hessian of the regularized function and showing it to be positive definite. Also, the iteration complexity result should follow from [42, 41] of the manuscript. Is my understanding correct? If so, any particular reason for choosing the constraint version (6) instead of the regularized version? If my understanding is not correct, what are the differences of the iteration complexity results with that of [42, 41] apart from those in Lines 191 - 193. This is crucial as in the absence of it, Section 2 needs to be toned down. Theorem 3.5 is interesting. Is the regularized version of (12) not geodesically convex? ======= After rebuttal ===== I appreciate the author rebuttal. It clarifies some confusions.

Reviewer 2



This approach to density matching is new to me. The paper is rather well motivated and written but some parts need clarification. - The main part of the paper consists in two parts: FR and KL ambiguity sets: * These parts seem (are?) somehow disconnected in terms of algorithmics. Could Pb. (12) be solved by (non-convex) gradient descent? * Given some accuracy could you compare the wallclock time between both approaches? * Could you precise the statement of line 59 on the statistical differences between FR and KL? * More generally, could you give (maybe as a table) a synthetic comparison between FR and KL approaches. - Could you provide more intuition about Algorithm 1: * why you need some kind of "ergodic geodesic average" in you algorithm. (See e.g. line 170-171) * Theorem 2.5 could be more emphasized, notably on how it relates to the complexity Theorem 2.7. - One of the drawback of the FR distance is that there is no closed form in the case where both mean and covariance are subject to ambiguity (footnote 1). However, the extent of the limitation is not clear: * How does this impact the KL part (section 3)? * In the application of Section 4 both classes means and covariances are different (225-226), so what approach to adopt? - Section 4/5 lack clarity compared to the rest of the paper. * 5.1 does not really bring much intuition about the behavior of the algorithm, it could be postponed to the supplementary * Section 4 and 5.2 could be merged, but 4 needs to be clarified (notably in relation with the question above). * The experimental setup could be more described (datasets, stopping criteria for FQDA and KQDA, etc.) Minor comments: * 109: By "solvable", do you mean feasible? More generally, could you provide a clear proof of Lemma 2.2 and clarify the paragraph 111-115 that I did not really get. * There is no condition present to ensure that the ambiguity sets do not intersect. Could this pose a problem?

Reviewer 3



computationally intractable" -- What is the argument or reference to support the proposed logical implication? Section 2, 3rd line: the Fisher information matrix itself may be rank deficient, hence only positive semidefinite rather than definite. Is the statement here that, once restricted to the tangent space, it is always positive definite? Eq. (6): perhaps recall what \hat \Sigma is here. Footnote 2: "the circle ... has positive sectional curvature": This isn't correct. The circle is a one-dimensional manifold, and as such it has zero (intrinsic) curvature. The footnote can be adapted if we make it to be a statement about the unit sphere S^2 in R^3. More globally, it is a bit tricky to make this statement about geodesic convexity of subsets of a compact manifold, when Definition 2.3 is specifically crafted for Hadamard manifolds (as indicated by the comment that precedes it). Depending on how exactly one adapts Def. 2.3, certains subsets of the sphere may or may not be deemed geodesically convex (and that may or may not be useful). Line 173: "closed form ... highly efficient": There are still a number of non-trivial computations involved, namely, matrix exponentials and logarithms. I expect a number of readers may find that this isn't "highly efficient", since such operations cost O(n^3) flops. In fact, you say as much on line 212. Line 181: There -> The Lines 190++, "An important difference is that [41, Theorem 9] requires the objective function to be Lipschitz continuous. Unfortunately, it seems difficult to establish whether L() satisfies this condition. We managed to circumvent the Lipschitz condition by proving that the Riemannian gradient of L() is bounded uniformly" -- I would expect that if the gradient is bounded (which is necessarily the case by compactness of the domain and continuity of the gradient), then the cost function is Lipschitz. Is that not the case here? What happens if we take X, Y two points in the set, and (since it is geodesically convex), we consider the unique geodesic connecting them, still in that set: let me call it c(t), with c(0) = X and c(1) = Y. Then, t -> f(c(t)) is a smooth function from R to R, hence we can apply the mean value theorem and claim there exists some t in (0, 1) such that f(Y) = f(X) + (f o c)'(t) = f(X) + . The gradient norm is bounded by G in the whole set, and c'(t) has norm equal to dist(X, Y) for all t since c is a geodesic, hence, by Cauchy-Schwarz: f(Y) - f(X) <= G dist(X, Y). Repeat with the roles of X and Y swapped, and we have found that f is Lipschitz continuous. Did I miss something?

[Author Response · NeurIPS 2019]

We would like to thank all referees for their appreciation of our results and the useful feedback. Below is our reply.

**Reviewer 1:** The regularized version of the FR problem is a geodesically convex optimization problem over the feasible set $\mathbb{S}_{++}^n$. However, the regularized problem has two drawbacks: *(i)* its objective function is not g-Lipschitz continuous over $\mathbb{S}_{++}^n$ because of the term $\langle S, \Sigma^{-1} \rangle$, *(ii)* $\mathbb{S}_{++}^n$ has infinite diameter. Due to these obstacles, the algorithms in [42, 41] cannot be used to solve the regularized problem. To the best of our knowledge there is no algorithm in the literature which can be readily applied to solve the regularized problem with convergence guarantee. By constraining the feasible set to $\mathcal{B}^{\mathrm{FR}}$, we can overcome these technical difficulties to establish the convergence guarantee in Theorem 2.7.

The KL divergence (confined to the subspace of Gaussian distributions) is not induced by any Riemannian metric. If we view problem (12) as a manifold optimization problem with the same Riemannian geometry as in Sect. 2, then (12) and its regularized version are both geodesically convex problems. However, problem (12) and its regularized version are convex in the usual Euclidean sense under the reparametrization $X = \Sigma^{-1}$, and it is more efficient to solve problem (12) by applying Theorem 3.2. Empirically, for dimension $d = 100$, on average solving the KL problem (12) using Theorem 3.2 takes $< 0.1$ seconds, while solving the FR problem (6) takes 1 second using Algorithm 1.

**Reviewer 2:** Your main suggestions for improvement focus around three aspects of the manuscript:

*1. Connection between FR and KL:* We apologize for not motivating thoroughly why we study both FR and KL. Ideally, we would like to use the FR metric since it is the unique metric that possesses the powerful invariance properties discussed in eqs. (4) and (5). These properties imply, amongst others, that the FR metric is invariant to the coordinate basis that frequently needs to be chosen arbitrarily in geometric problems. While failing to satisfy these desirable properties, the KL divergence constitutes an approximation to the FR metric (as discussed in footnote 1 of the appendix) that is computationally more tractable. We propose to elaborate on these connections in the introduction. To further illustrate the commonalities and differences between the two approaches, we also propose to replace Section 5.1 (which, as you correctly pointed out, does not add much insight) with a section that visualizes and compares the decision boundaries of the nominal QDA and those of FR and KL in the context of our application. In particular, we observe that our approaches lead to non-hyperbolic decision boundaries in general. We will also add a comparison of the wallclock times in Section 5.2. As we pointed out in our response to Rev. 1, solving the KL problem (12) using Theorem 3.2 takes $< 0.1$ seconds on average for dimension $d = 100$, while solving the FR problem (6) takes 1 second using Algorithm 1. Because (12) is non-convex, the gradient descent algorithm cannot guarantee to converge to global minimum of (12).

*2. Further explanations for Sections 4+5:* Thank you for pointing out the lack of explanation in the main paper regarding the ambiguity set used in this application. Appendix A of the manuscript argues that for a fixed sample size, estimating $\hat{\Sigma}_c$ is much harder than estimating $\hat{\mu}_c$. In our numerical experiments, we thus identify $\hat{\mu}_c$ with the sample average and only consider uncertainty in $\hat{\Sigma}_c$. We will add a discussion in the paper that summarizes our findings from the appendix and clarifies which ambiguity set we use.

*3. Intuition for the use of the ergodic geodesic average in Algorithm 1:* Thank you for pointing out this omission. The ergodic geodesic average $\bar{\Sigma}_k$ is the sequence that has been proven to converge in [42], and we are not aware of any last-iterate convergence results under the similar conditions of problem (6). We propose to clarify this aspect in the camera-ready version of the paper.

Thank you also for your minor suggestions, which we plan to address in the revised version of the manuscript.

**Reviewer 3:** We apologize for the lack of rigor in our use of the term "computationally intractable". We meant to say that the problem is non-convex (since the $L^2$-Wasserstein manifold of Gaussian measures has a non-negative sectional curvature (see [36]), and the objective function is not geodesically convex on this manifold) and therefore *appears* to be computationally intractable, but we do not have a rigorous hardness result. We will fix this in the revised version. We also agree that the Fisher information matrix may be rank deficient if we have a degenerate Gaussian distribution, in which case the inner product on the tangent space would fail to be positive definite. Since we work with the set $\mathcal{M}$ of all *non-degenerate* Gaussian distributions (cf. line 34 of the paper), however, the Fisher information matrix will always be positive definite. We will highlight this in the revised version of the paper. As for footnote 2, thank you for pointing out that the circle has zero intrinsic curvature; we will update the paper accordingly. In line 173, we will replace the statements "closed form" and "highly efficient" with the appropriate complexity estimates. The proof of Theorem 9 in [42] involves bounding the gradient via using the Lipschitz assumption. For this argument to be valid, the Lipschitz assumption needs to hold on an open subset $\mathcal{Y} \subseteq \mathcal{M}$ containing $\mathcal{B}^{\mathrm{FR}}$. We simplified the proof a little bit by directly bounding the gradient on $\mathcal{B}^{\mathrm{FR}}$ which can be done for our problem. Regarding linear convergence rate: under the (minor) assumption that $S \succ 0$, we can show that the objective function is strongly g-convex and g-smooth over the ball $\mathcal{B}^{\mathrm{FR}}$ (the explicit constants can be computed from $\lambda_{\min}(S)$, $\lambda_{\max}(S)$ and the bounds in Lemma C.1). Theorem 15 in [42] applies, and Algorithm 1 (with an appropriately modified stepsize) converges linearly for the sequence $\Sigma_k$. We will add this result in the revised version. Empirically, we observe the linear convergence rate even when $S$ is singular. Thank you very much for your suggestion!

[Meta-Review · NeurIPS 2019]

The paper seems well-written and motivated. Overall evaluations are positive, but the authors are strongly encouraged to take the concerns raised into account. Also, adding a compact ball as a constraint leads to a projection based method, where the authors should present results about the extra computational costs incurred due to that.